# Cloning of the wheat leaf rust resistance gene *Lr47* introgressed from *Aegilops speltoides*

Hongna Li [1,8], Lei Hua [1,8], Shuqing Zhao[2,8], Ming Hao [3], Rui Song [1], Shuyong Pang[1,2], Yanna Liu[1], Hong Chen[3], Wenjun Zhang[4], Tao Shen [1], Jin-Ying Gou [5], Hailiang Mao[6], Guiping Wang [1], Xiaohua Hao [1], Jian Li[1], Baoxing Song [1], Caixia Lan[6], Zaifeng Li[2], Xing Wang Deng [1], Jorge Dubcovsky [4,7], Xiaodong Wang [2] ✉ & Shisheng Chen [1] ✉

Leaf rust, caused by *Puccinia triticina* Eriksson (*Pt*), is one of the most severe foliar diseases of wheat. Breeding for leaf rust resistance is a practical and sustainable method to control this devastating disease. Here, we report the identification of *Lr47*, a broadly effective leaf rust resistance gene introgressed into wheat from *Aegilops speltoides*. *Lr47* encodes a coiled-coil nucleotide-binding leucine-rich repeat protein that is both necessary and sufficient to confer *Pt* resistance, as demonstrated by loss-of-function mutations and transgenic complementation. *Lr47* introgression lines with no or reduced linkage drag are generated using the *Pairing homoeologous1* mutation, and a diagnostic molecular marker for *Lr47* is developed. The coiled-coil domain of the Lr47 protein is unable to induce cell death, nor does it have self-protein interaction. The cloning of *Lr47* expands the number of leaf rust resistance genes that can be incorporated into multigene transgenic cassettes to control this devastating disease.

Wheat is one of the leading food crops providing one-fifth of the total food calories and proteins consumed by humans. Reducing yield losses inflicted by fungal pathogens is an effective measure to increase wheat production. *Puccinia triticina* Eriksson (*Pt*), the causal agent of wheat leaf rust (or brown rust), is one of the most devastating fungal pathogens in wheat. This disease occurs in most wheat growing regions and can cause significant yield losses in susceptible wheat varieties under favorable climatic conditions[1,2]. Breeding for leaf rust resistance is considered the most practical and sustainable way to control this devastating disease.

To date, over 80 leaf rust resistance (*Lr*) genes have been officially designated in wheat and its wild relatives[3]. However, due to the size and complexity of the wheat genome, only ten *Lr* genes have been cloned so far either by classical map-based cloning (*Lr1, Lr10, Lr21, Lr34, Lr42,* and *Lr67*)[4–6] or by rapid gene-cloning approaches including MutRenSeq (*Lr13*), TACCA (*Lr22a*), MutChromSeq (*Lr14a*), and MutIsoSeq (*Lr9*)[7–11]. Among the cloned *Lr* genes, *Lr34* and *Lr67* are known as slow rusting genes encoding a putative ATP-binding cassette transporter and a hexose transporter, respectively[4,5]. *Lr14a* encodes a protein containing 12 ankyrin repeats, and *Lr9* encodes a protein with an N-terminal tandem kinase domain followed by vWA/Vwaint domains at its C-terminus[9,10]. The other six isolated genes have been shown to encode typical coiled-coil nucleotide-binding leucine-rich repeat (NLR) proteins[6–8,11].

[1]National Key Laboratory of Wheat Improvement, Peking University Institute of Advanced Agricultural Sciences, Shandong Laboratory of Advanced Agricultural Sciences in Weifang, 261325 Shandong, China. [2]State Key Laboratory of North China Crop Improvement and Regulation, College of Plant Protection, Hebei Agricultural University, 071000 Baoding, Hebei, China. [3]Triticeae Research Institute, Sichuan Agricultural University, 611130 Chengdu, China. [4]Department of Plant Sciences, University of California, Davis, CA 95616, USA. [5]Key Laboratory of Crop Heterosis and Utilization (MOE) and Beijing Key Laboratory of Crop Genetic Improvement, China Agricultural University, 100193 Beijing, China. [6]National Key Laboratory of Crop Genetic Improvement, Huazhong Agricultural University, 430070 Wuhan, China. [7]Howard Hughes Medical Institute, Chevy Chase, MD 20815, USA. [8]These authors contributed equally: Hongna Li, Lei Hua, Shuqing Zhao. ✉e-mail: zhbwxd@hebau.edu.cn; shisheng.chen@pku-iaas.edu.cn

Cloning of additional *Lr* genes is desirable to diversify the combinations of *Pt* resistance genes used in transgenic cassettes or gene pyramids to achieve durable resistance. The diploid wheat species *Aegilops speltoides* Tausch ($2n = 2x = 14$, SS), the closest extant donor of the B-genome to bread wheat[12,13], harbors valuable *Lr* genes, such as *Lr28*, *Lr35*, *Lr36*, *Lr47*, *Lr51*, and *Lr66*[14]. These genes have been successfully transferred from *Ae. speltoides* to common wheat, but none of them has been cloned so far due to the limited recombination observed between wheat and *Ae. speltoides* chromosomes.

The all-stage resistance gene *Lr47* was transferred from chromosome 7S#1 of *Ae. speltoides* to the short arm of chromosome 7A of the hexaploid wheat translocation line T7AS·7S#1S·7AS·7AL using homoeologous recombination in the presence of the *Pairing homoeologous1* (*ph1b*) mutation[13], but the ancestral origin of this *Ae. speltoides* segment is not clear[15,16]. Using C-banding and restriction fragment length polymorphism (RFLP) markers, it was determined that the genetic length of the translocated segment 7S#1S was approximately 20–30 centimorgans (cM) long and that it was located 2–10 cM from the centromere[13]. In a recent study, the length of the *Ae. speltoides* segment 7S#1S was estimated to be between 157 and 174 Mb (Chinese Spring RefSeq v1.0 coordinates) based on a set of simple sequence repeat (SSR) markers[17]. This interstitial translocation segment carrying *Lr47* was subsequently backcrossed into several spring wheat cultivars, such as Pavon, Express, Kern, RSI5, Yecora Rojo, and UC1041[16,18]. However, the presence of the *Lr47* introgression was found to be associated with several detrimental effects on agronomic and quality traits, including reduced grain yield, decreased flour yield, and increased flour ash[16]. Therefore, new rounds of homoeologous recombination using the *ph1b* mutation are needed to reduce the size of the introgressed segment containing *Lr47* and minimize linkage drag. The linkage between *Lr47* and these negative effects can also be broken by cloning the gene and generating transgenic plants. This is a worthwhile endeavor because *Lr47* is one of a few genes known to confer strong levels of resistance against a wide range of *Pt* isolates[1,13,17,19–23], and may play an important role in the improvement of wheat resistance to leaf rust.

Here, we report the identification of the *Lr47* gene through a combination of map-based cloning, ethyl methanesulfonate (EMS) mutagenesis and transcriptome sequencing (MutRNASeq)[24] approaches. We also validate the function of the cloned candidate gene using independent susceptible EMS mutants, barley stripe mosaic virus (BSMV)-mediated gene editing[25], and stable transgenic complementation. To minimize potential linkage drag, we generate smaller *Ae. speltoides* chromosome segments carrying *Lr47* using the *ph1b* mutation. Finally, we develop a diagnostic molecular marker to accelerate the deployment of *Lr47* in wheat breeding programs.

## Results

### *Lr47* provides broadly effective resistance to leaf rust
The bread wheat line Kern *Lr47* (PI 603918*7/Kern) is near isogenic to wheat cultivar Kern and carries the introgressed *Ae. speltoides* chromosome segment 7S#1S. In the seedling tests, all Kern *Lr47* plants showed strong resistance to *Pt* race THDB, whereas all Kern plants were susceptible (Fig. 1a). In a subset of 128 $F_2$ plants from the cross Kern *Lr47* × Zhengzhou5389 (ZZ5389, susceptible control) evaluated with race THDB, 93 plants were resistant and 35 were susceptible, which fits a 3:1 segregation ratio expected for a single dominant genetic locus ($\chi^2 = 0.38$, $P = 0.54$). To quantify the growth of *Pt* pathogen at 2, 4, 6, and 8 days post inoculation (dpi) in Kern *Lr47* and its recurrent parent Kern, we measured the average infection areas using fluorescence microscopy and the fluorescent dye WGA-FITC, which stains the *Pt* pathogen. At all four time points, the average infection areas observed microscopically were significantly smaller ($P < 0.001$) in plants with *Lr47* than in those without the gene (Fig. 1b, c). Fluorescent images of *Pt* growth in Kern showed a diffuse

network of fungal hyphae at the edges of the infected sites, indicating that *Pt* appeared to spread unimpeded. In contrast, pathogen growth in Kern *Lr47* was restricted and no significant differences were observed in the fungal infection areas from 4 to 8 dpi (Fig. 1c). Microscopic examination of leaf samples from Kern *Lr47* also revealed the accumulation of phenolic auto-fluorogens at each infection site, starting from 2 dpi, and showing a substantial increase at 4 dpi (Supplementary Fig. 1).

Seedlings of *Lr47* near-isogenic lines (NILs; Express *Lr47*, UC1041 *Lr47*, and RSI5 *Lr47*) and their recurrent parents (Express, UC1041, and RSI5) were challenged with 23 additional *Pt* races collected in China (Supplementary Table 1). All *Lr47* NILs exhibited strong levels of resistance [infection types (ITs) = 0 to ;] to the tested races (Fig. 1d and Supplementary Table 1). In contrast, the recurrent parents showed a wide range of responses to different *Pt* races, probably due to other *Lr* genes present in these genetic backgrounds. Nevertheless, inoculation of these recurrent parents revealed that at least one accession was fully susceptible to each *Pt* race (Fig. 1d), confirming that *Lr47* confers resistance to these *Pt* races.

### Characterization of the *Ae. speltoides* segment carrying *Lr47*
To determine the physical location and size of the *Ae. speltoides* chromosome segment introgressed into hexaploid wheat, we compared the single nucleotide polymorphisms (SNPs) identified in the RNA sequencing (RNA-seq) of *Lr47* NILs (Kern *Lr47*, Yecora Rojo *Lr47*, and UC1041 *Lr47*) with those from another ten sequenced hexaploid wheat varieties and three *Ae. speltoides* accessions (AE915, AE1590, and PI 554292; Supplementary Data 1). We focused on the SNPs that were present in the three *Ae. speltoides* accessions, but absent in the ten sequenced hexaploid wheat varieties, and that are hereafter referred to as *Ae. speltoides*-specific SNPs. Based on this rule, we identified 3,169 *Ae. speltoides*-specific SNPs (Supplementary Data 2) that were shared with *Lr47* NILs in the proximal region of chromosome arm 7AS starting from 40.4 Mb to 190.5 Mb (CS RefSeq v1.1 coordinates, Supplementary Fig. 2). The translocation breakpoints ranged from 40,050,116 bp to 40,398,277 bp at the terminal end, and from 190,476,279 bp to 191,833,539 bp at the proximal end (Supplementary Data 2). Thus, the *Ae. speltoides* segment in the three *Lr47* NILs ranges from 150.1 Mb to 151.8 Mb.

To better define the translocation breakpoints, we designed 16 7A/7S genome-specific primers across the ~150 Mb introgressed *Ae. speltoides* segment (Supplementary Data 3), and used them to genotype the *Lr47* NILs and their recurrent parents. We found that translocation breakpoints in the *Lr47* NILs were located between markers *pku0738* and *pku0745* at the terminal end, and between markers *pku2216* and *pku2233* at the proximal end (Supplementary Fig. 3).

### Fine mapping of *Lr47* using *ph1b*-induced homoeologous recombination
No recombination was expected between wheat chromosome 7A and the introgressed 7S chromosome segment in the presence of the *Ph1* gene. Genotyping of 128 $F_2$ plants from the cross Kern *Lr47* × ZZ5389 using the flanking markers *pku0745* and *pku2216* (Supplementary Data 3) revealed no recombination between these two markers, confirming that recombination between *Ae. speltoides* and *T. aestivum* chromosomes is suppressed. To induce recombination between the 7A/7S homoeologous chromosomes, the introgression line Kern *Lr47* was crossed with the CS *ph1b* mutant (CS*ph1b*). Supplementary Fig. 4 describes the procedure for identifying recombinants between the 7S and 7A chromosomes. A total of 15 $F_2$ plants heterozygous for *Lr47* and homozygous for *ph1b* were obtained and self-pollinated to generate $F_3$ seeds for the recombination screening. We genotyped 2654 plants from eight selected segregating $F_3$ families with the flanking markers *pku0745* and *pku2216*, and identified 542 plants with recombination events.

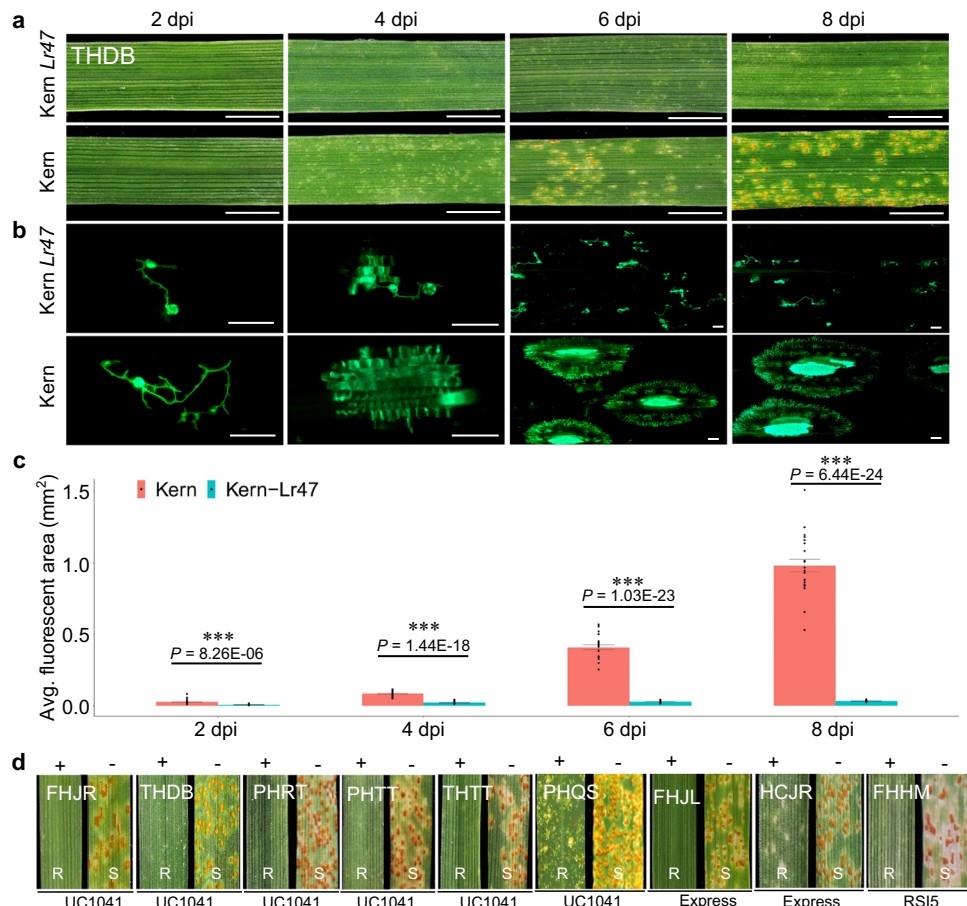

**Fig. 1 | Resistance responses of leaf rust resistance gene *Lr47*. a** Infection types in the introgression line Kern *Lr47* and its recurrent parent Kern in response to *Pt* race THDB at 2, 4, 6, and 8 dpi. Scale bars represent 0.5 cm. **b** *Pt* infection areas visualized by WGA-FITC staining. Leaves were collected at 2, 4, 6, and 8 dpi, cleared with KOH, and stained with WGA-FITC. Scale bars represent 100 μm. **c** Interaction graphs showing the average size of individual fungal infection areas of Kern *Lr47* (turquoise blue) and Kern (red) estimated by fluorescence microscopy at 2, 4, 6, and 8 dpi. Data were collected from at least 20 independent fungal infection sites ($n \geq 20$). Black dots represent single data points. Asterisks indicate the level of significance by two-sided unpaired *t* test. ***$P < 0.001$. Error bars are standard errors of the means. **d** Infection types of *Lr47* NILs and their recurrent parents (UC1041, Express, and RSI5) in response to selected *Pt* races FHJR, THDB, PHRT, PHTT, THTT, PHQS, FHJL, HCJR, and FHHM. +, resistant *Lr47* allele present; −, no resistant *Lr47* allele. R resistant, S susceptible. Source data are provided as a Source data file.

The recombinants were genotyped using another 13 7A/7S-genome specific markers across the introgressed *Ae. speltoides* chromosome segment (Fig. 2a, b). Among the recombinants, we only selected plants heterozygous for a smaller *Ae. speltoides* chromosome segment for phenotyping. Finally, we identified *Ae. speltoides* segments of seven different lengths, which were designated as L1 to L7 (Fig. 2c). The selected recombinants were evaluated for resistance to race THDB at the three-leaf stage in growth chambers. Plants carrying introgressions L1 and L4 were susceptible, so *Lr47* was mapped to a 3.5-Mb region (CS RefSeq v1.1 coordinates) flanked by markers *pku1104* and *pku1152* (Fig. 2c). The candidate gene region includes 49 annotated high-confidence genes in Chinese Spring (*TraesCS7A02G110400–-TraesCS7A02G115200*, Supplementary Data 4). The functional annotation of these genes revealed six typical NLR genes (Supplementary Data 4), a gene class frequently associated with disease resistance in plants[26,27].

### Identification of the *Lr47* candidate gene by modified MutRNASeq

Since the recurrent parent Kern was susceptible to *Pt* races THDB and PHQS (Supplementary Fig. 5), these two races were subsequently used to identify susceptible EMS mutants of Kern *Lr47*. Among the progenies of 662 M₂ mutant families screened with *Pt* race THDB, we identified ten independent families with susceptible plants and

confirmed their susceptibility to races THDB and PHQS by evaluating M₃ seeds derived from the susceptible plants (Fig. 3a and Supplementary Fig. 6a). Genotyping using six 7A/7S-genome specific markers confirmed that all the susceptible mutant lines carried the complete *Ae. speltopides* chromosome segment.

To refine the mapping of *Lr47* in the Kern *Lr47* × CS*ph1b* population, we crossed one of the obtained susceptible EMS mutant lines, m118, with the non-mutagenized resistant Kern *Lr47*, and generated an F₂ population consisting of 1,141 individuals. We performed whole genome re-sequencing of the parental lines, identified EMS-induced SNPs, and generated sequence-based markers in the *Lr47* candidate region. We used these markers to genotype lines with recombination events in the target region and phenotyped these lines for resistance to *Pt*. Based on these results, we mapped *Lr47* between markers *pkus675* and *pkus175* (Fig. 2d), within the same interval as in the Kern *Lr47* × CS*ph1b* population. The reduced candidate region corresponds to a physical interval of 2.5 Mb in the reference genome of the *Ae. speltoides* accession "TS01" (Fig. 2e). Within the candidate gene region in the TS01 genome, we found six typical NLR genes (Fig. 2e and Supplementary Data 5).

To identify the candidate gene of *Lr47*, we first generated RNA-seq reads (≥69.8 million 150-bp paired-end reads per sample) from *Pt*-inoculated leaves of Kern *Lr47* and ten independent susceptible M₃ mutants (Supplementary Table 2). We used inoculated leaves because

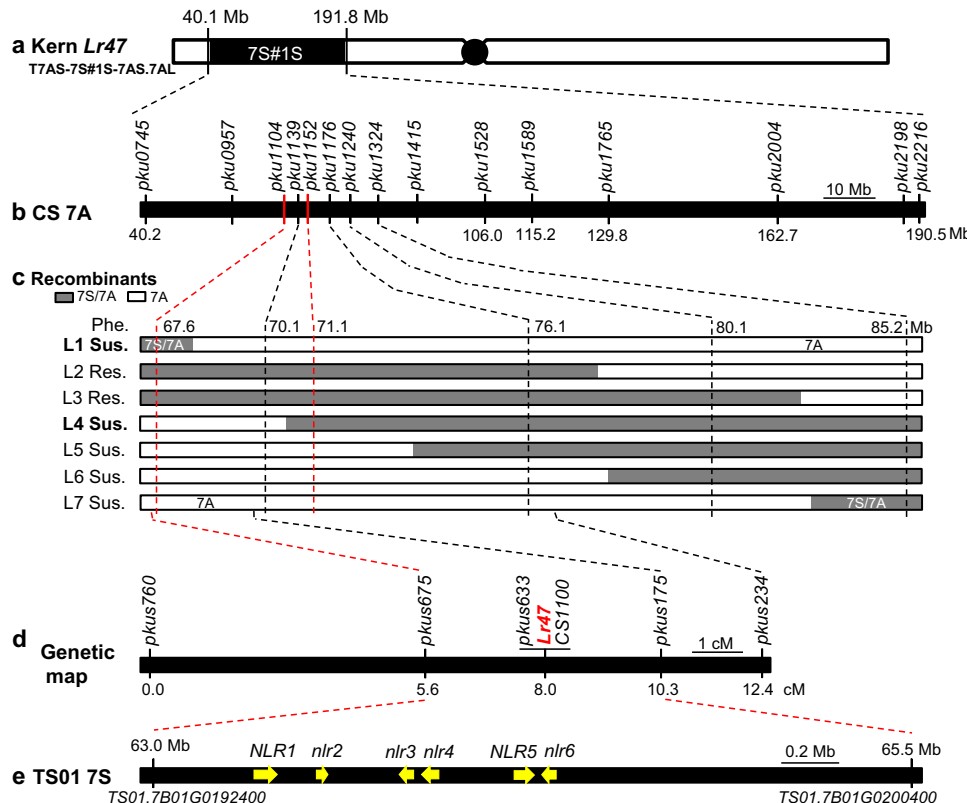

**Fig. 2 | Fine mapping of the leaf rust resistance gene *Lr47*. a** Schematic representation of recombinant chromosome T7AS-7S#1S-7AS.7AL in Kern *Lr47* carrying the introgressed *Ae. speltoides* segment 7S#1S (indicated by a black rectangle). **b** Fine mapping of *Lr47* using 7A/7S homoeologous recombinants induced by the *ph1b* mutant. The genome-specific markers (Supplementary Data 3) distributed along the introgressed 7S#1S chromosome segment and their physical locations based on the Chinese Spring reference genome RefSeq v1.1. **c** Recombinant haplotypes (L1–L7) with *Ae. speltoides* segments of different lengths determined by marker analysis. White rectangles represent chromosome 7A; gray rectangles indicate heterozygous for 7S/7A; Sus. susceptible, Res. resistant. **d** Genetic map of *Lr47* based on 1141 F$_2$ plants from cross m118 × Kern *Lr47*. **e** Physical map of *Lr47* in the sequenced reference genome of *Ae. speltoides* TS01. NLR genes are represented by yellow arrows and upper-case letters (*NLR*) and pseudogenes by lower-case letters (*nlr*).

we did not know if the *Lr47* gene is induced by the pathogen or not. Assembly of the Kern *Lr47* sequenced reads yielded 146,715 high confidence transcript contigs (≥500 bp). We performed BLASTN searches of the Kern *Lr47* transcriptome database using the sequences of the candidate NLR genes in CS and TS01 as queries, and obtained 45 transcript contigs with a BLAST *e*-value = 0. We mapped the sequenced reads of ten susceptible mutants against the 45 selected Kern *Lr47* transcript contigs and found one contig named KN638873_g379_i12 with EMS-type (G/C-to-A/T) point mutations in all ten mutants (Supplementary Figs. 6b and 7a). Using the primer pairs *EMS8054F2R1*, *EMS8054F1R4*, and *EMS8054F7R7* (Supplementary Data 3) developed from this contig, we performed PCRs to amplify the region containing the mutations and confirmed the presence of nucleotide transitions in these susceptible mutants (Supplementary Fig. 7b). In addition, genotyping of the critical recombinants derived from the cross m118 × Kern *Lr47* with marker *EMS8054F1R4* confirmed that the C-to-T SNP in m118 co-segregated with the disease phenotype.

To determine the structure of the *Lr47* candidate gene in contig KN638873_g379_i12, we (i) assembled the whole genome resequencing reads to obtain a genomic sequence containing the *Lr47* candidate; (ii) confirmed the transcript and genomic sequences of the *Lr47* candidate by PCR and Sanger sequencing; and (iii) compared the transcript with the corresponding genomic sequence. These analyses revealed that the *Lr47* candidate gene has six exons encoding a typical NLR protein of 928 amino acids, which is hereafter referred to as *CNL2* (Fig. 3b; GenBank accession number OQ919262). Using 5′ and 3′ rapid amplification of cDNA ends (RACE; Supplementary Fig. 8), we determined that the 5′-untranslated region (UTR) of *CNL2* is 1142 bp long

with three introns, and the 3′UTR is 607 bp long with only one intron (Fig. 3b). The predicted 2787 bp coding sequence of *CNL2* is 95.6% and 91.9% identical to its closest homologs within the candidate regions in *Ae. speltoides* TS01 and CS, respectively. Based on the predicted coding sequence, we found that four of the EMS mutations introduced premature stop codons and the others introduced nonsynonymous amino acid changes (Fig. 3c). AlphaFold prediction of the full-length CNL2 protein yielded a structural model with the expected structures for the coiled-coil (CC) and nucleotide-binding site (NB) domains as well as a leucine-rich repeat (LRR) domain containing 17 repeat units forming a typical α/β horseshoe fold (Supplementary Fig. 9).

**Validation of *CNL2* using BSMV-sgRNA-based gene editing**
Because the BSMV-sgRNA-based gene editing method requires a wheat line expressing the *Cas9* gene, we first crossed the introgression line Kern *Lr47* with the cultivar Bobwhite, which carries a highly expressed *Cas9* allele[25]. The resulting F$_1$ plants, carrying both the *Lr47* and *Cas9* genes, were inoculated with BSMV-sgRNA constructs targeting the candidate gene *CNL2*. Leaves of the inoculated F$_1$ plants showed chlorotic spots and white streaks, consistent with BSMV-induced viral symptoms (Supplementary Fig. 10). Next-generation sequencing of amplicons prepared with DNA isolated from the infected leaves of F$_1$ plants showed a somatic editing efficiency of ~16%.

Infected F$_1$ (M$_0$) plants were self-pollinated to produce M$_1$ seeds. Genotyping of 216 M$_1$ plants identified 24 plants (11.1%) with heterozygous or homozygous mutations. Among them, two plants (mut-1 and mut-2) that were homozygous for "T" or "TT" deletions at position 2,280 downstream of the ATG in the cDNA (Fig. 4a) were selected for

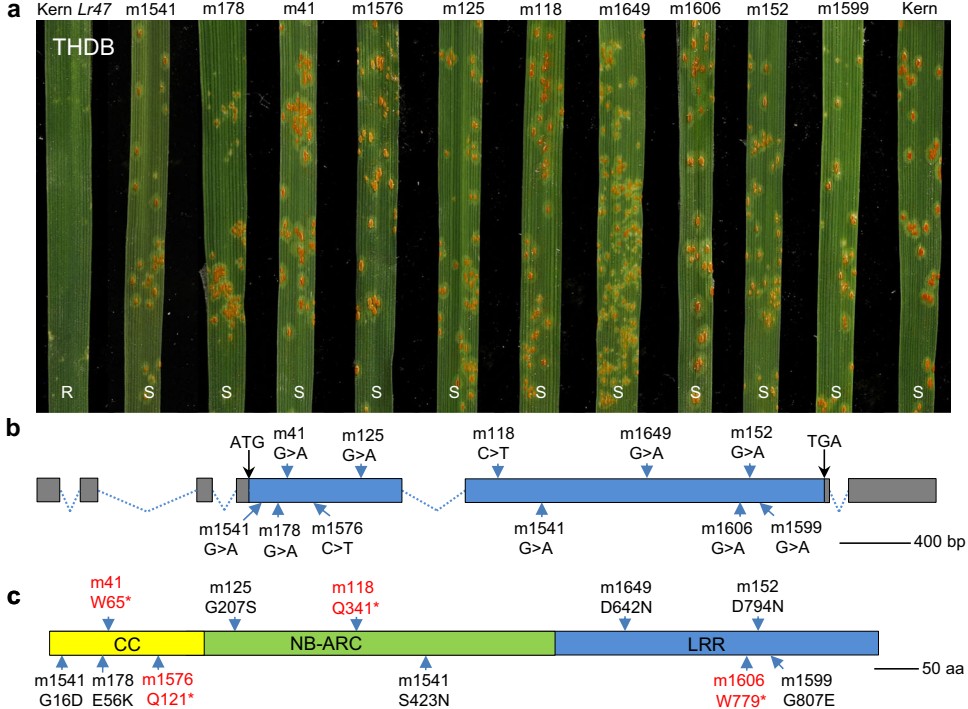

**Fig. 3 | Validation of the *Lr47* candidate gene using EMS mutants. a** Susceptible EMS mutants used to isolate the *Lr47* candidate gene. Infection types for Kern *Lr47* (positive control), ten independent EMS mutants, and Kern (negative control) inoculated with *Pt* race THDB. R resistant, S susceptible. **b** Gene structure of the *Lr47* candidate gene. The positions of the EMS-derived loss-of-function mutations are indicated by blue arrows. The mutant line m1541 carries two mutations. Gray boxes indicate untranslated regions, blue boxes represent coding exons, and dotted lines represent introns. **c** Protein structure of Lr47, with the predicted amino acid changes caused by the EMS mutations indicated. Mutation names in red indicate premature stop codons and in black indicate amino acid substitutions. CC domain, NB domain, and LRR motifs are shaded with yellow, green, and blue, respectively. Source data are provided as a Source data file.

further characterization. These frameshift deletions alter 18% of the protein sequence, resulting in loss of function of the CNL2 protein. Evaluations of the progeny of the two selected plants with race THDB showed that the plants homozygous for the deletions displayed susceptible reactions similar to those of *Cas9*-transgenic Bobwhite, whereas Kern *Lr47* and its sister line without editing were resistant (Fig. 4b). These results indicate that *CNL2* is required for *Lr47*-mediated resistance to *Pt*.

**Wheat plants transformed with *CNL2* were resistant to leaf rust**

To determine whether *CNL2* is sufficient to confer resistance to leaf rust, a 7,234 bp genomic DNA fragment from the introgression line Kern *Lr47*, including the complete coding region, introns, and regulatory sequences (Fig. 4c), was transformed into the spring wheat cultivar Fielder via *Agrobacterium tumefaciens*-mediated transformation. A total of 80 independent transgenic $T_0$ plants were generated, and the presence of the transgene was confirmed with the primer pairs *EMS8OS4F7R7* and *Lr47speF5R5* (Supplementary Data 3), which amplify the transcribed region of *CNL2*. All transgenic plants containing the transgene and Kern *Lr47* (positive control) showed high levels of resistance to *Pt* race PHQS, whereas the untransformed control Fielder was completely susceptible (Supplementary Figs. 11a and 12). Among the transgenic plants, we found plants with both lower and higher levels of resistance than Kern *Lr47* (Supplementary Figs. 11a and 12), suggesting that these transgenic plants may have different numbers of *CNL2* insertions or expression levels.

To test this hypothesis, the qRT-PCR primer pair *Lr47qPCRF2R3* was used to assess the transcript levels of *CNL2* in 11 randomly selected transgenic $T_0$ plants ($T_0$CS401-1 to $T_0$CS401-11) using wheat *ACTIN* gene as an endogenous control. *CNL2* transcript levels were significantly higher in all selected transgenic $T_0$ plants than in the Fielder control ($P < 0.01$), and eight of them showed higher *CNL2* transcript levels than Kern *Lr47* ($P < 0.05$; Supplementary Fig. 11b). Approximately 25 $T_1$ plants from each selected transgenic family were genotyped with the primer pair *Lr47speF5R5*, and all except three showed significant departures from the expected segregation ratio of 3:1 (transgenic/non-transgenic), with an excess of transgenic plants (Supplementary Table 3). The number of *CNL2* insertions was determined by Taq-Man copy number assays (Supplementary Table 3). Overall, plants derived from three of the transgenic events ($T_1$CS401-2, $T_1$CS401-7, and $T_1$CS401-8) were estimated to have only a single copy of the transgene, whereas the other eight transgenic families were estimated to have between two and five *CNL2* copies (Supplementary Table 3). The estimated *CNL2* copy number based on TaqMan assays correlated significantly with the transcript levels in the selected transgenic plants ($R = 0.90$, $P < 0.001$).

Transgenic $T_1$ plants from the 11 selected transgenic events were challenged with seven *Pt* races, all of which are virulent on Fielder. All plants from the $T_1$ transgenic families $T_1$CS401-1 and $T_1$CS401-10, which were fixed for the transgene, exhibited a high level of resistance, and resistance in the transgenic families $T_1$CS401-2 and $T_1$CS401-7 perfectly co-segregated with the presence of the transgene (Fig. 4d and Supplementary Fig. 13). The more effective resistance observed in transgenic families $T_1$CS401-1 and $T_1$CS401-10 reflected the higher number of *CNL2* insertions (or higher expression levels) present in them. These results demonstrated that *CNL2* is sufficient to confer resistance to all *Pt* races tested (Fig. 4d). Inoculation of transgenic plants and the Fielder control with race THDB did not provide useful information because Fielder was highly resistant (ITs = 1-) to this *Pt* race (Supplementary Fig. 14).

Taken together, the high-resolution genetic map, loss-of-function EMS and BSMV-*Cas9*-induced mutations, and the transgenic results demonstrated that the candidate *CNL2* is *Lr47*.

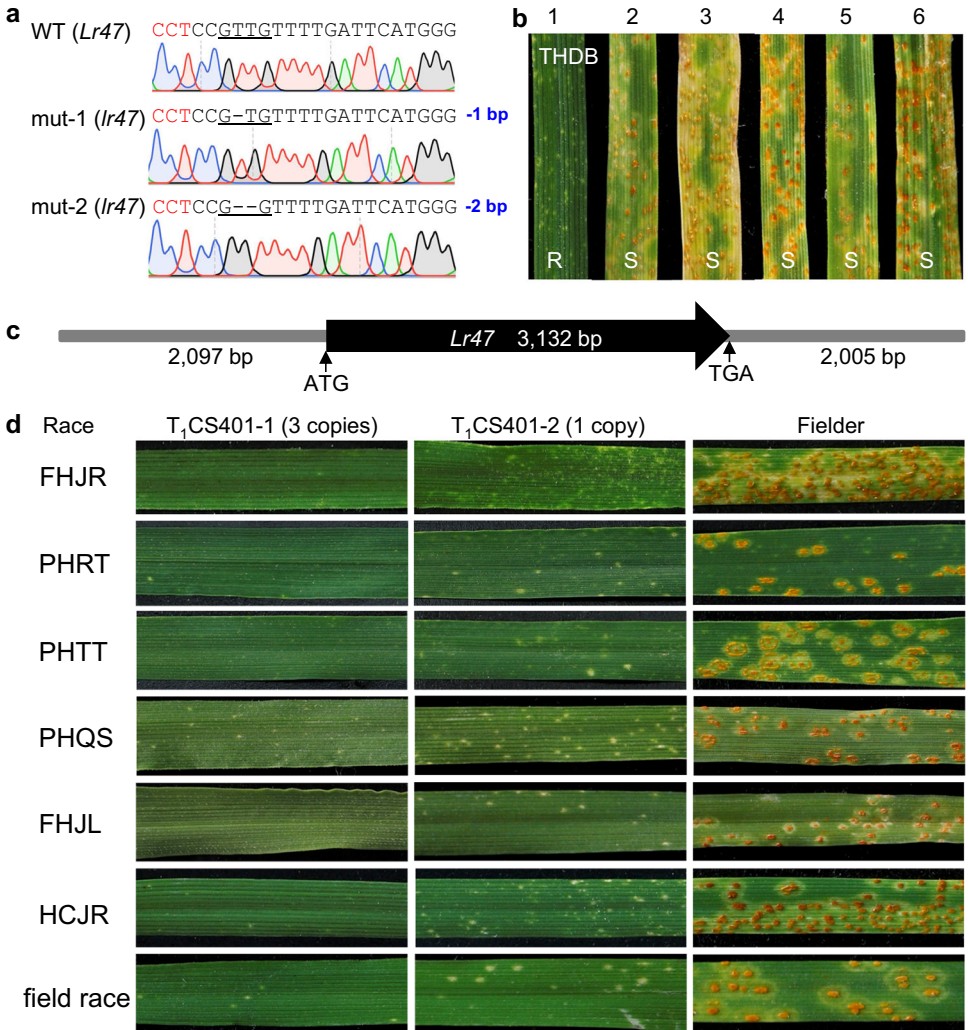

**Fig. 4 | Functional validation of *Lr47* by BSMV-mediated gene editing and transgenic complementation. a** Sequencing chromatogram showing the induced polymorphisms between the wild type (WT) and the selected BSMV-*Cas9*-induced editing mutants. The mutated nucleotides are underlined and the CCT PAM sequence is highlighted in red. **b** Reactions to *Pt* race THDB. (1) Kern *Lr47*; (2, 3) mutant line mut-1; (4, 5) mutant line mut-2; and (6) *Cas9*-transgenic Bobwhite. **c** The 7,234-bp genomic DNA fragment including *Lr47* used for transformation. The black arrow indicates the *Lr47* gene (from ATG to TGA). **d** Infection types of two transgenic families T₁CS401-1 and T₁CS401-2 and Fielder control in response to *Pt* races FHJR, PHRT, PHTT, PHQS, FHJL, HCJR, and a mixture of naturally prevalent *Pt* races collected from the field in 2021 (field race, 36°26′04.0″N, 119°26′42.6″E). Copy number of transgenes was estimated using TaqMan assays (Supplementary Table 3). R resistant, S susceptible. Source data are provided as a Source data file.

## The diagnostic marker and the origin of the *Lr47* segment

A search of the released reference genomes identified the closest homologs of *Lr47* on chromosomes 7A, 7B, and 7D of diploid, tetraploid, and hexaploid wheat, as well as on chromosome 7 of *Sitopsis* species (including *Ae. speltoides*, *Ae. longissima*, *Ae. sharonensis*, and *Ae. searsii*), which had 83.29–92.78% protein similarity to Lr47. The phylogenetic tree in Supplementary Fig. 15 shows that Lr47 and its homologs from *Sitopsis* species are in the same phylogenetic clade. Multiple alignment of the protein sequences revealed that most of the variation was present in the LRR domain (Supplementary Fig. 16).

A dominant marker, *Lr47mas*, was designed based on the DNA polymorphism C1059G (resulting in amino acid change D353E), which distinguishes *Lr47* from all other homologs found so far (Supplementary Fig. 16). PCR amplification with the marker *Lr47mas* (Supplementary Data 3) yielded an amplicon of 488 bp only when *Lr47* was present (Supplementary Fig. 17). Using this marker, we examined a large collection of wheat genotypes, including 144 accessions of *T. aestivum*, 78 of *T. turgidum*, 24 of *T. monococcum*, and 118 of *Ae. speltoides* (Supplementary Table 4). PCR amplicons of the predicted size were present in only three of the *Ae. speltoides* accessions but were absent in all diploid, tetraploid, and hexaploid wheat accessions (except for six *Lr47* introgression lines) tested in this study (Supplementary Table 4). To confirm the presence of *Lr47* in the three *Ae. speltoides* lines (T2140002, Y162, and Y397), we designed two pairs of gene-specific primers, *Lr47SHF3R7* and *Lr47SHF2R1* (Supplementary Data 3), which amplify the complete coding region of *Lr47*. Sanger sequencing confirmed that these three *Ae. speltoides* accessions carry a functional *Lr47* allele identical to that of Kern *Lr47*.

*Ae. speltoides* includes two subspecies: *Ae. speltoides* var. *speltoides* and *Ae. speltoides* var. *ligustica*. Since the taxonomic subspecies and origin of the *Lr47* introgressed segment are not clear[15,16], we compared the SNPs identified from Kern *Lr47* RNA-seq data with those of eight *Ae. speltoides* var. *speltoides* accessions, six *Ae. speltoides* var. *ligustica* lines, and two *Ae. speltoides* lines for which the subspecies is unknown but carry *Lr47* (T2140002 and Y162). We focused only on the polymorphisms that were within the ~150 Mb introgressed segment 7S#1S and were polymorphic among the 16 *Ae. speltoides* accessions described above, but were absent in the recurrent parent Kern. Based

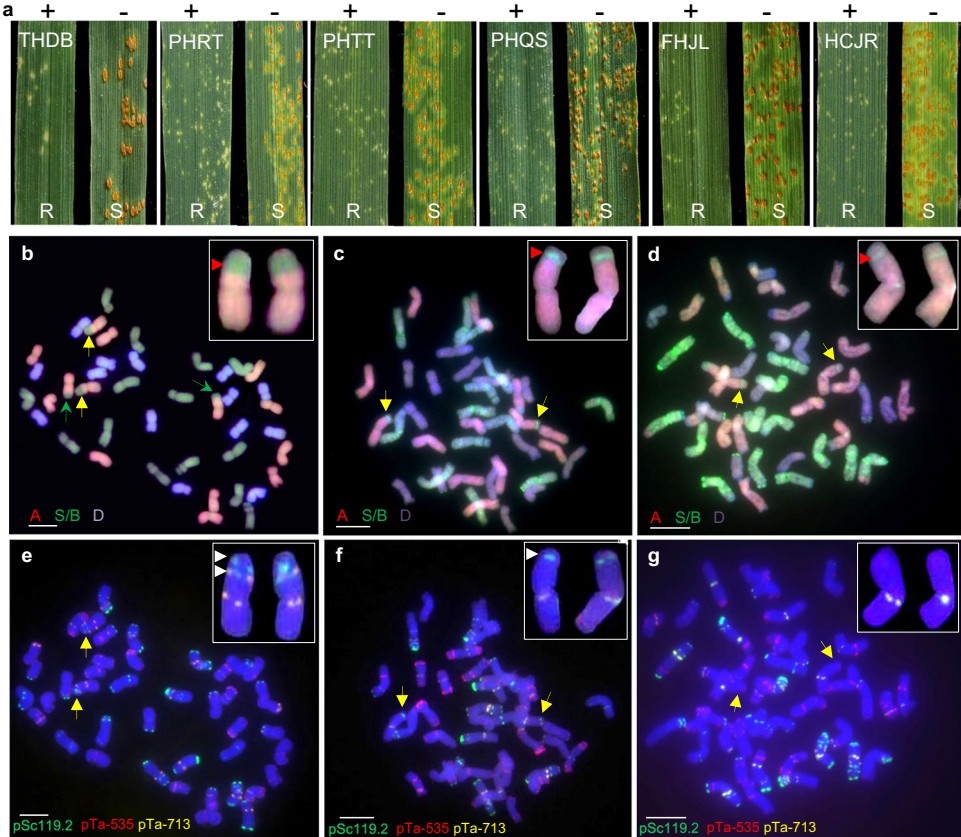

**Fig. 5 | Reduction in the length of the introgressed *Ae. speltoides* chromosome segment carrying *Lr47*. a** Infection types of homozygous $BC_1F_2$ plants from the introgression line YM21 + *Lr47*-2 (+) and the plants lacking *Lr47* (-). Six *Pt* races (THDB, PHRT, PHTT, PHQS, FHJL, and HCJR) were used for evaluation. R, Resistant; S, Susceptible. **b–d** GISH images of wheat lines Kern *Lr47*, L2 ($BC_2F_2$), and L8 ($BC_1F_2$). The magnified images show the *Ae. speltoides* chromosome segments introgressed into wheat chromosome 7A (*Ae. speltoides* chromatin is painted in green and marked with red arrowheads). **e–g** FISH images from wheat lines Kern *Lr47*, L2 ($BC_2F_2$), and L8 ($BC_1F_2$). Probes pSc119.2 (green), pTa535 (red), and pTa713 (yellow) were used. Yellow arrows indicate the wheat-*Ae. speltoides* translocated 7A chromosomes, and magnified images of these chromosomes are shown in the insets. Green arrows indicate the 4AL/7BS translocation. Scale bar = 10 µm. All experiments were repeated three times independently with consistent results. Source data are provided as a Source data file.

on this approach, we obtained 199 polymorphic sites (Supplementary Data 6). A Neighbor-Joining tree based on these polymorphisms showed that Kern *Lr47* is located in a branch encompassing multiple *Ae. speltoides* var. *speltoides* accessions (Supplementary Fig. 18), suggesting that the translocated segment originated from this subspecies. The two accessions that most closely resemble to Kern *Lr47* are T2140002 and Y162, which carry *Lr47*. The geographical origins of T2140002 and Y397 (carrying *Lr47*) are unknown, but the most closely related accession Y162 was collected in Iraq, suggesting that Iraq and its neighboring countries are a likely region where *Lr47* originated.

**Reducing the length of the introgressed segment carrying *Lr47***

Because the large introgressed *Ae. speltoides* segment 7S#1S was associated with several undesirable traits, we reduced the size of the introgressed segment carrying *Lr47* using the *ph1b* mutation (Supplementary Fig. 19). We started from recombinant L2 (Fig. 2c), which has a smaller introgression segment of approximately 40 Mb with a recombination event between 76.1 and 80.1 Mb (Fig. 2c and Supplementary Fig. 20). Subsequently, the L2 plant was self-pollinated and its progeny was used for a second round of recombination with the *ph1b* mutant. Using the markers *pku0745* and *pku1176*, we identified 20 recombinants from 313 segregating individuals. These recombinants were genotyped with the 7A/7S genome-specific markers from this region (Supplementary Data 3). Among them, we identified one critical recombinant, which we hereafter refer to as F284-128 (L8). The crossover breakpoint in L8 was between markers *pku0957* (58.8 Mb)

and *pku1026* (63.0 Mb), so the 7S chromosome segment in this resistant line is reduced to between 13.1 Mb and 21.3 Mb (Supplementary Fig. 20d).

Recombinant L8 with the truncated 7S segment was crossed and backcrossed once with the Chinese common wheat variety Yangmai21 (YM21), which is susceptible to multiple *Pt* races, including THDB, PHRT, PHTT, PHQS, FHJL, and HCJR. Four PCR markers, *pku1026*, *Lr47speF5R5*, *Lr47mas*, and *pku1176* (Supplementary Data 3), were used to confirm the presence of the truncated 7S segment in the selected $BC_1F_2$ plants (hereafter referred to as YM21 + *Lr47*-2). Homozygous $BC_1F_2$ plants from the introgression line YM21 + *Lr47*-2 challenged with six different Chinese *Pt* races showed high levels of resistance (ITs = 0 to 1), whereas the recurrent parent YM21 and the sister line lacking the smaller alien chromosome segment exhibited susceptible infection types (ITs = 3 to 4) in response to the same races (Fig. 5a).

We then performed genomic in situ hybridization (GISH) experiments to validate the results of the molecular marker analysis described above. GISH analyses confirmed the presence of the interstitial *Ae. speltoides* segment in Kern *Lr47* on the recombinant chromosome arm 7A/7S (Fig. 5b), whereas recombinants L2 and L8 had significantly smaller alien chromosome segments (Fig. 5c, d and Supplementary Fig. 21). Kern *Lr47* also carries the previously reported 4AL/7BS intergenomic translocation (Fig. 5b), which is present in both durum and bread wheat[28]. Moreover, we observed that the *ph1b* mutant induced additional recombination between the homoeologues of the A, B, and D subgenomes (Fig. 5d). Those translocations will be eliminated when

the reduced introgressed segment is backcrossed into the target commercial wheat varieties.

Fluorescence in situ hybridization (FISH)-based karyotype analysis revealed strong Oligo-pSc119.2 and Oligo-pTa713 signals on the introgressed *Ae. speltoides* chromosome segment 7S#1S in Kern *Lr47* (Fig. 5e), whereas only the pSc119.2 signal was detected in L2 (Fig. 5f), suggesting that the proximal region of the introgressed segment 7S#1S was eliminated by the recombination event in L2 (Fig. 2c). In L8, neither Oligo-pSc119.2 nor Oligo-pTa713 signals were detected (Fig. 5g and Supplementary Fig. 21b), suggesting that the distal region of the alien chromatin was also replaced by wheat chromatin after the second round of homoeologous recombination. These results are consistent with the analysis with PCR markers (Supplementary Fig. 20) and with the GISH results (Fig. 5b–d).

### Agronomic and quality evaluation of the generated *Lr47* introgression line

To evaluate the effects of the truncated alien chromosome segment containing *Lr47* (35.9–40.0 Mb, Supplementary Fig. 20c) on agronomic and quality traits, recombinant L2 was crossed and backcrossed three times with the hexaploid wheat variety YM21 and self-pollinated to generate the $BC_3F_3$ seeds (YM21 + *Lr47*-1; Supplementary Fig. 19). This line was selected because it was available earlier than L8. We planted $BC_3F_3$ sister lines homozygous for the presence or absence of the alien chromosome segment in both the greenhouse and growth chamber, and measured the phenotypic changes. Under disease-free conditions, no significant differences were observed between the developed introgression line YM21 + *Lr47*-1 and its sister line for 15 morphological and quality traits examined (Fig. 6). The only exception was the spike length (SL), which was significantly shorter in YM21 + *Lr47*-1 than in its sister line ($P < 0.05$; Fig. 6c). The results were consistent between the growth chamber and greenhouse experiments (Supplementary Fig. 22).

In a field experiment in Shandong Province, China, the introgression line YM21 + *Lr47*-1 was highly resistant to *Pt*, whereas the sister control line was fully susceptible (Fig. 6q, r). Analysis of morphological traits showed that SL was also significantly shorter in YM21 + *Lr47*-1 than in the isogenic sister line without the *Lr47* introgression (Fig. 6s).

### Characterization of *Lr47* encoding an NLR immune receptor protein

Transcript levels of *Lr47* relative to *ACTIN* were measured in Kern *Lr47* by qRT-PCR. We observed no significant transcriptional differences between plants inoculated with *Pt* race THDB and mock inoculated with water at 1, 2, 4, and 6 dpi (Fig. 7a), indicating that *Lr47* is not induced by the presence of the *Pt* pathogen.

The predicted Lr47 protein includes an N-terminal CC domain, a central NB site, and a C-terminal LRR region, with two predicted monopartite nuclear localization signals determined by the cNLS Mapper program (amino acids 22–54 and 533–539; Fig. 7b). To determine the subcellular localization of Lr47, a green fluorescent protein (GFP) tag was used to visualize Lr47 in tobacco (*Nicotiana benthamiana*) leaves and wheat protoplasts. Both cytoplasmic and nuclear fluorescence were observed in tobacco leaves for all constructs, namely GFP-Lr47_CDS (coding region), Lr47_CDS-GFP, GFP-Lr47_CC, and Lr47_CC-GFP constructs (Fig. 7c). Anti-GFP immunoblots confirmed that the proteins were expressed at the expected sizes for all constructs (Fig. 7d). In wheat protoplasts, cytoplasmic and nuclear signals were also detected in GFP-Lr47_CDS, Lr47_CC-GFP, and GFP-Lr47_CC constructs (Supplementary Fig. 23).

To test whether the full length Lr47 and/or the CC domain alone are capable of triggering cell death in *N. benthamiana*, we investigated signaling after *Agrobacterium*-mediated transient expression in leaves of *N. benthamiana*. We did not observe cell death or obvious yellowing in leaf regions transiently expressing Lr47 and its protein domains individually. In contrast, robust cell death was observed in leaf regions expressing BAX (an inducer of cell death)[29] and the CC domain of Sr13[30], which were used as positive controls (Fig. 7e and Supplementary Fig. 24). Moreover, the self-interaction ability of Lr47 was initially assessed using yeast two-hybrid assays. We did not detect any direct interaction between the Lr47 protein itself and its protein domains CC, NB, and LRR in vitro (Supplementary Fig. 25).

## Discussion

Wild relatives of wheat are valuable sources of disease resistance genes, and they have been used previously for introgressing *Lr* genes into common wheat varieties[3]. However, cloning resistance genes in wheat relatives is often hampered by suppressed recombination in the regions carrying alien introgressions. Several recently developed technologies, such as MutRenSeq, MutRNASeq, MutChromSeq, and MutIsoSeq, have facilitated the isolation of resistance (*R*) genes in wheat and its wild relatives[10,24,31]. Here, we report the cloning of the broadly effective leaf rust resistance gene *Lr47* using a combination of *ph1b*-induced recombination and the MutRNASeq method. Using this approach, the *CNL2* candidate gene for *Lr47* was successfully identified. The susceptibility to *Pt* races of the mutants induced by EMS or BSMV-mediated gene editing and the resistance of the *CNL2* transgenic Fielder lines (Figs. 3 and 4) confirmed that this candidate gene is both necessary and sufficient to confer resistance to leaf rust, and therefore that *CNL2* is *Lr47*.

MutRNASeq is a valuable approach to identify candidate genes by comparing the selected EMS-induced susceptible mutants with the wild type using RNA-seq data[24]. Compared with the previously reported approach[24], the modified MutRNASeq developed in this study (Supplementary Fig. 6b) does not require the generation of a physical map/reference sequence across the map interval. This feature is useful and may spark the interest of researchers in identifying target genes in recombination-sparse regions, especially in plant species with large genomes. It is worth noting that RNA-seq might be biased by tissue or sampling time, so alternative sequencing strategies like whole genome re-sequencing or exome capture sequencing might be preferable. In this study, the wild-type Kern *Lr47* was sequenced at very high sequencing depth (>250 million reads, Supplementary Table 2) on the Illumina NovaSeq 6000 platform to ensure a high-quality transcriptome assembly. *Lr47* was found within a complex locus comprising a cluster of NLR genes that exhibit both copy number and structural variations (Supplementary Data 4 and 5). Since NLR genes are the most common class of genes associated with disease resistance in wheat and other plants[26,27], we hypothesized that one of the NLR genes might be *Lr47* and therefore prioritized NLR genes within the candidate region (Fig. 2d, e) for further analysis. This significantly reduced the amount of sequencing data to be analyzed.

Recently, virus-based sgRNA delivery systems have been developed and tested in wheat and other plant species[25,32]. Compared with the biolistic-based or *Agrobacterium*-mediated gene editing systems, the virus-based delivery systems do not require conventional plant genetic transformation and regeneration procedures[25]. In our study, the BSMV-based sgRNA delivery system was able to generate both somatic and heritable mutations in wheat via infection of previously obtained *Cas9*-transgenic wheat plants[25,33]. We demonstrated that the *Cas9*-transgenic wheat plants can be crossed with a line carrying the targeted alien chromosome (Kern *Lr47*) to generate edits in the $F_1$ plants (Fig. 4a, b). The results suggest that the BSMV-based sgRNA delivery system could reduce genotype dependency for targeted gene editing by transferring the high-expressing *Cas9* locus into different wheat cultivars. Considering the simplicity of the BSMV-sgRNA infection procedure, this genome editing tool has great potential for applications in wheat.

One limitation for the incorporation of alien segments by homologous recombination is that the introgressed chromosome segments

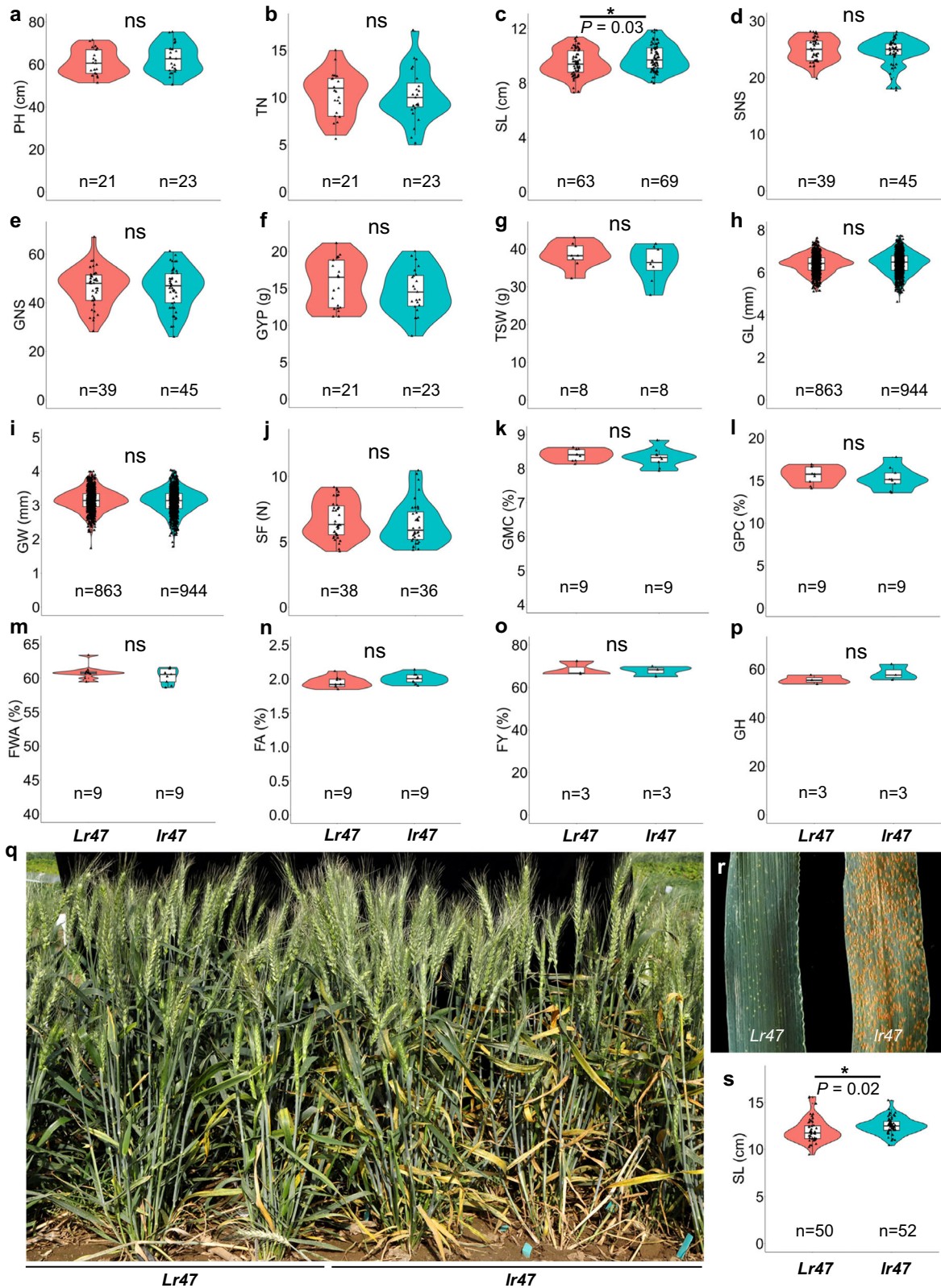

are usually large and can carry linked genes with negative agronomic or quality effects. Examples of linkage drag of alien introgressions carrying resistance genes include *Lr19* from *Thinopyrum ponticum*, *Pch1* from *Ae. ventricosa*, and *Pm16* from *Ae. speltoides*[34,35]. Similarly, the introgression of the *Ae. speltoides* segment 7S#1S carrying *Lr47* was reported to be associated with several negative effects on agronomic and quality traits[24]. To enhance the utility of *Lr47* in wheat breeding, we

attempted to eliminate the deleterious linkage drag by two rounds of *ph1b*-induced homoeologous recombination. Using a combination of marker analysis, cytology, and *Lr* phenotypic screening (Fig. 5), two recombinants with much smaller *Ae. speltoides* chromosome segments (L2: ~36 Mb and L8: ~13 Mb) were identified.

A preliminary characterization of the introgression line YM21 + *Lr47*-1 in greenhouse and chamber experiments showed no significant

**Fig. 6 | Statistical analysis of agronomic and quality traits in YM21 + *Lr47*-1 and its sister control line. a–p** Plants grown in a controlled walk-in growth chamber at 24 °C day/22 °C night with a 16 h light/8 h dark photoperiod were phenotyped for the following traits: **a** plant height (PH); **b** tillers number (TN); **c** spike length (SL); **d** spikelet number per spike (SNS); **e** grain number per spike (GNS); **f** grain yield per plant (GYP); **g** thousand-seed weight (TSW); **h** grain length (GL); **i** grain width (GW); **j** shearing force (SF) in Newtons (N); **k** grain moisture content (GMC); **l** grain protein content (GPC); **m** flour water absorption (FWA); **n** flour ash (FA); **o** flour yield (FY); and **p** grain hardness (GH). **q–s** Plants grown in the field. **q** Visual phenotypes of YM21 + *Lr47*-1 and its sister line in the field. **r** Infection types of YM21 + *Lr47*-1 and the sister control line. **s** Spike length (SL). In each box plot, the horizontal line corresponds to the median score, box edges show the first and third quartiles, and whiskers extend to the maximum and minimum values within 1.5 times the interquartile range. The shape of the violin plot reflects the distribution of the variable. Black triangles represent single data points. The significance of the differences was estimated using two-sided unpaired *t* test. ns not significant ($P > 0.05$), *$P < 0.05$. *Lr47*, resistant *Lr47* allele present (YM21 + *Lr47*-1); *lr47*, sister control line lacking the alien chromosome segment. Source data are provided as a Source data file.

adverse effects on morphological and quality traits in the absence of disease. The only exception was SL, which was significantly shorter in YM21 + *Lr47*-1 than in its isogenic sister line (Fig. 6c). Shorter spikes were also observed in the field experiment (Fig. 6s). Despite reducing SL, the presence of the *Lr47* introgression was not associated with reduction in spike grain yield (Fig. 6 and Supplementary Fig. 22). We are currently introgressing the shorter ~13 Mb *Ae. speltoides* chromosome segment into the Chinese common wheat cultivar YM21 to determine whether the shorter spikes observed in YM21 + *Lr47*-1 is due to linkage with other alien genes or pleiotropic effects of the *Lr47* gene. The latter is unlikely because no significant differences were observed in SL between transgenic plants homozygous for the transgene (1 copy) and the untransformed control Fielder under disease-free conditions (Supplementary Fig. 26). If necessary, the flanking markers and the diagnostic marker for *Lr47* (Supplementary Data 3) can be used to develop wheat lines with even smaller introgressed segments carrying the *Lr47* gene.

For NLR proteins in plants, the CC domain is thought to play a crucial role in downstream signaling. Previous studies reported that the CC domains of numerous NLR proteins (such as MLA10, Pm21, Sr33, and Sr50) are sufficient to cause cell death after transient expression in *N. benthamiana* leaves[36,37]. However, other NLR proteins, including RPM1, RPS5, Rx, and Bs2, have CC domains that do not trigger cell death in leaves of *N. benthamiana*[36,37]. In this study, we found that the CC domain of the Lr47 protein cannot directly induce cell death (Fig. 7e and Supplementary Fig. 24), nor does it have self-protein interaction (Supplementary Fig. 25), indicating that an unknown activation mechanism is essential for Lr47 function. Although NLR proteins have been shown to mediate hypersensitive response by forming pentameric disease resistosomes in *Arabidopsis* (ZAR1)[38] and wheat (Sr35)[39], the activation and downstream pathways of resistance for Lr47 require further investigation.

*Lr47* provides strong resistance to virulent *Pt* races in a wide range of regions, including the United States[17], Mexico[1], Russia[23], South Africa[23], the Indian sub-continent[22,40], and China (the current study). Leaf rust isolates virulent on *Lr47* have not been identified. However, elucidation of the typical NLR structure of *Lr47* indicates that its resistance may not be durable in the field because of rapid changes in leaf rust pathogen populations. The rust pathogen may evolve through the mutation or deletion of the corresponding *Avr* genes, leading to the breakdown of resistance conferred by NLR genes[41,42]. For instance, the Ug99 race group of the wheat stem rust pathogen (*Puccinia graminis* f. sp. *tritici*) has successfully broken the resistance of wheat varieties carrying the resistance genes *Sr24*, *Sr31*, *Sr36*, *Sr38*, and *SrTmp*[43,44]. Therefore, a combination of *Lr47* with other slow-rusting multi-pathogen resistance genes, such as *Lr34/Sr57/Yr18*[4] and *Lr67/Sr55/Yr46*[5], is a preferred strategy for developing wheat cultivars with durable resistance against this devastating rust pathogen.

In conclusion, the identification of *Lr47*, the generated introgression lines with eliminated or reduced deleterious linkage drag, and the available diagnostic marker developed in this study provide useful tools to diversify the deployed *Lr* genes and accelerate the use of *Lr47* in modern wheat breeding programs.

## Methods

### Plant materials and mapping populations
The six pairs of hard red spring wheat NILs with and without the leaf rust resistance gene *Lr47*[13,16] used in this study are listed in Supplementary Table 5. A total of 7590 recombinant gametes from two mapping populations were used to construct a high-density genetic map of *Lr47*. The first one included 2654 plants from selected F$_3$ families that were homozygous for the *ph1b* mutation and segregating for the *Ae. speltoides* chromosome segment (population Kern *Lr47* × CS*ph1b*). The second population, which included 1,141 F$_2$ individuals, was generated from a cross between a Kern *Lr47* EMS mutant susceptible to *Pt* designated m118 and the wild type (Kern *Lr47*). Since both parents (m118 and Kern *Lr47*) carry the same *Ae. speltoides* chromosome, recombination is normal in the presence of the *Ph1b* wild-type allele. Moreover, Kern *Lr47* was crossed with the susceptible wheat line ZZ5389 to characterize leaf rust resistance in the presence of the *Ph1* gene. A collection of 118 accessions of *Ae. speltoides*, 24 of *T. monococcum*, 78 of *T. turgidum*, and 144 of *T. aestivum* (including six *T. aestivum* genetic stocks carrying the *Lr47 Ae. speltoides* segment 7S#1S) was used to determine the value of the diagnostic marker developed in this study for marker assisted selection (Supplementary Table 4).

### Leaf rust assays and pathogen growth in infected leaves
Leaf rust seedling evaluations were performed at the Peking University Institute of Advanced Agricultural Sciences, Weifang, China and Hebei Agricultural University, Baoding, China. A total of 24 *Pt* races and their avirulence/virulence profiles are presented in Supplementary Table 1. The *Pt* races THT-, PHT-, THJ-, and PHJ- were predominant in China and showed virulence to most of the designated *Lr* genes[45]. Seedlings at the three-leaf stage were inoculated with fresh *Pt* urediniospores mixed with talcum powder at a ratio of 1:20 using the shaking off method[46]. The inoculated plants were placed in a dark dew chamber set at 22 °C for approximately 24 h and then maintained at 22–24 °C with a 16 h photoperiod. ITs of plants were scored at ~12 dpi using a 0–4 scale[47].

To estimate the growth of the *Pt* pathogen and cell death triggered by *Lr47* in NILs, leaf segments (~6 cm long) from Kern *Lr47* and its recurrent parent Kern inoculated with race THDB were sampled at 2, 4, 6, and 8 dpi. Following bleaching with a 1:1 mixture of ethanol and glacial acetic acid, and subsequent transparency using chloral hydrate, the direct observation of phenolic auto-fluorogens' accumulation at each infection site is possible under epi-fluorescence microscopy. To observe the fungal structures, the collected leaves were autoclaved in 1 M KOH, stained with wheat germ agglutinin labeled with fluorescein isothiocyanate (WGA-FITC; Cat No. L4895-10MG, Sigma-Aldrich, USA), and visualized by a Zeiss Discovery V20 fluorescence dissecting microscope (Zeiss, Jena, Germany)[30,48].

### EMS mutant screening
Approximately 20,000 seeds of wheat line Kern *Lr47* were treated in 16 beakers, each containing 250 mL of 0.8% EMS (Cat No. M0880-25G, Sigma-Aldrich, USA). The beakers were incubated for 18 h at 25 °C on a shaker at 150 rpm. Next, the treated seeds were washed five times with 300 mL tap water and then placed under running water for 3 h. All surviving M$_1$ plants were grown in a greenhouse, and a total of 4,756

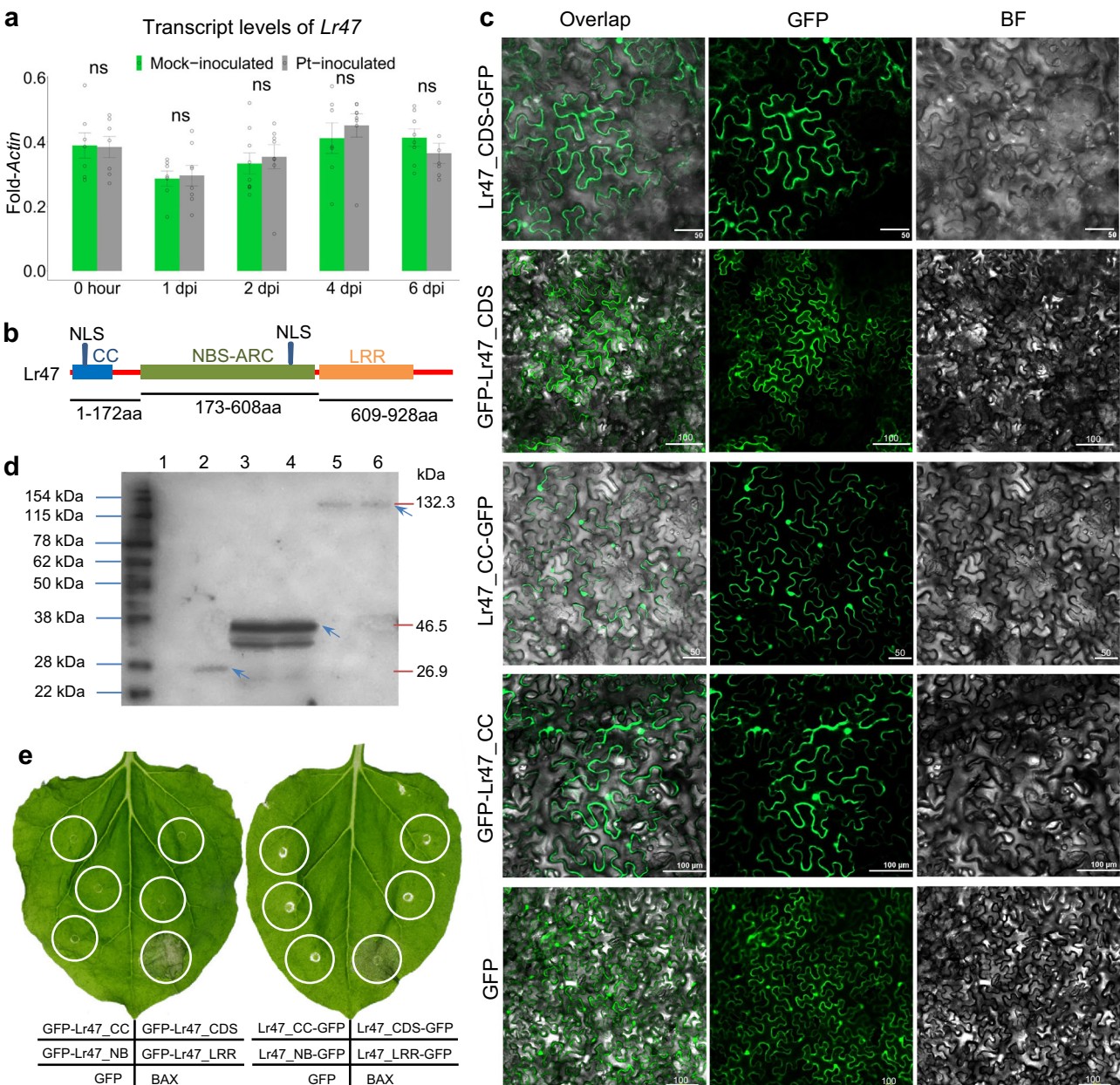

**Fig. 7 | Characterization of Lr47. a** Transcript levels of *Lr47* in mock-inoculated and *Pt*-inoculated plants. Leaves were collected at five time points: 0 h, 1 dpi, 2 dpi, 4 dpi, and 6 dpi. Transcript levels were determined in at least seven biological replicates (n ≥ 7) and expressed as fold-*Actin*. Gray open dots represent single data points. The significance of the differences was estimated using two-sided unpaired *t*-test. Error bars indicate SEMs. ns = not significant. **b** Schematic diagram of conserved domains in the Lr47 protein. NLS, the predicted nuclear localization signal. **c** Subcellular localization of the GFP-fused Lr47 protein in tobacco leaves. This experiment was repeated three times with consistent results. Scale bars represent 50 or 100 μm. BF, bright field; GFP, green fluorescent protein. **d** GFP-horseradish

peroxidase Western blot showing proteins expressed at expected sizes for all constructs. Target bands are highlighted by blue arrowheads. 1, empty vector; 2, GFP; 3, GFP-Lr47_CC; 4, Lr47_CC-GFP; 5, Lr47_CDS-GFP; 6, GFP-Lr47_CDS. The experiment was repeated twice with consistent results. **e** Macroscopic cell death in *N. benthamiana* leaves 48 hpi with *A. tumefaciens* carrying Lr47 constructs. No cell death was observed in leaf regions transiently overexpressing Lr47 and its protein domains individually. CC coiled-coil, NB nucleotide binding, LRR leucine-rich repeat, CDS the coding regions of Lr47, BAX a mammalian cell death inducer as positive control. Source data are provided as a Source data file.

independent M₂ families were obtained. Kern *Lr47* and its recurrent parent Kern were evaluated with five *Pt* races, including THDB, PHQS, PHTT, PHRT, and FHJR. Approximately 25 M₂ seeds per family were challenged with *Pt* race THDB in growth chambers at the Peking University Institute of Advanced Agricultural Sciences, Weifang, China. The M₃ seeds derived from susceptible M₂ plants were re-evaluated with races PHQS and THDB. Finally, all the identified mutants were genotyped with 7S-specific primers to rule out the possibility of seed contamination.

## Sequences, markers, MutRNASeq, and bioinformatics analysis

RNA-seq of *Lr47* NILs, *Ae. speltoides* accessions (AE915, AE1590, and PI 554292), and the susceptible EMS mutant lines were performed at Beijing Novogene Bioinformatics Technology Co., Ltd. (Beijing, China). All the raw sequencing data was deposited at the National Genomics Data Center under BioProject accession number PRJCA016987. The released reference genomes of hexaploid wheat Chinese Spring (CS; RefSeq v1.1) and other nine wheat genotypes from the Wheat Pan Genome Project were used for comparative analysis[49,50]. RNA-seq data

of Yecora Rojo *Lr47*, UC1037 *Lr47*, and White Yecora *Lr47* were generated from a previous study[20]. Raw RNA-seq reads were trimmed using Trimmomatic software v0.32 to remove low-quality reads and adaptors[51]. Trimmed reads were aligned to the CS reference sequence using STAR software v2.7.10a[52]. Variant calls were made using Freebayes v1.3.6[53] and filtered using BCFtools v1.14 (https://github.com/samtools/bcftools).

The modified MutRNASeq was performed as follows: RNA-seq reads from Kern *Lr47* were assembled de novo using Spades version 3.14.1[54]. We performed BLASTN searches of the transcriptome database of Kern *Lr47* using sequences of typical NLR genes within the *Lr47* candidate regions from the reference genomes of CS and TS01[55] as queries, and obtained transcript contigs with similarity values greater than 85% to the homologs or paralogs within the interval that *Lr47* was mapped to. Next, the RNA-seq reads of the ten susceptible EMS mutants were mapped to these transcript contigs using BWA v0.7.12 and SAMtools v1.8[24].

To develop markers in the m118 × Kern *Lr47* population, we performed whole genome re-sequencing for both parents (accession number PRJCA016987). We aligned the m118 and Kern *Lr47* sequences of the genes in the candidate region and identified the polymorphic sites. Genome-specific primers were designed using the Primer 3 online software (v0.4.0, https://bioinfo.ut.ee/primer3-0.4.0/primer3/) and used to amplify regions carrying putative *Ae. speltoides*-specific SNPs. The detected SNPs were used to develop Insertion-Deletion (InDel) and cleaved amplified polymorphic sequence (CAPS) markers[56,57]. The other published reference genome of *Ae. speltoides* AEG-9674-1[55] was used to support marker development.

## 5' and 3' RACE

5'- and 3'-RACE reactions were carried out using total RNA extracted from leaves of the introgression line Kern *Lr47*. RACE reactions were performed using the Invitrogen FirstChoice RLM-RACE Kit (Cat. no. AM1700, ThermoFisher Scientific, MA, USA) according to the manufacturer's instructions. PCR products from the RACE reactions were cloned into the pMD18-T vector (TaKaRa, Kyoto, Japan) using the TA cloning method. The selected positive clones were sequenced using the Sanger method.

## BSMV-sgRNA-based gene editing

The BSMV-based guide RNA delivery system[25] was used to validate the *Lr47* candidate gene. The previously reported plasmids (pCB301-BSMVα, pCB301-BSMVβ, and pCB301-BSMVγ-sg)[25] for BSMV-sgRNA-based gene editing in wheat were used in our study. BSMV-sgRNA constructs were transformed into *A. tumefaciens* strain EHA105. *Agrobacterium* lines harboring the BSMV-derived plasmids expressing the sgRNA were mixed and co-infiltrated into *N. benthamiana* leaves[25]. Infiltrated *N. benthamiana* leaves were harvested and homogenized for wheat inoculation[25]. The wheat transgenic line exogenously expressing *Cas9* gene in the genetic background of Bobwhite (*Cas9*-transgenic Bobwhite) was crossed with the introgression line Kern *Lr47*, and the resulting $F_1$ plants were infected with BSMV vectors targeting the *Lr47* candidate gene. The infected $M_0$ plants were self-pollinated to produce $M_1$ seeds. We identified mutations in the $M_1$ plants using the primer pair *Lr47BSMVF2R1* (Supplementary Data 3). Finally, the edited plants and their progenies were challenged with *Pt* race THDB, which is virulent on Bobwhite.

## Wheat transformation and copy number assays

A 7,234-bp genomic DNA fragment consisting of the entire coding region and introns (3132 bp) of *Lr47*, 2097 bp upstream of the start codon, and 2005 bp downstream from the stop codon was amplified from the introgression line Kern *Lr47* by PCR using the PrimeStar Max DNA Polymerase (TaKaRa, Kyoto, Japan). The fragment was inserted into the linearized binary vector pCAMBIA1300 using an In-Fusion® HD

Cloning Kit (Clontech, CA, USA). The resulting plasmid pCAMBIA1300-*Lr47* was introduced into the susceptible hexaploid wheat line Fielder via *A. tumefaciens* (strain EHA105)-mediated transformation. Primer pairs *EMS8054F7R7* and *Lr47speF5R5* (Supplementary Data 3) were used to confirm the presence of the transgene, and the qRT-PCR primer pair *Lr47qPCRF2R3* (Supplementary Data 3) was used to estimate the transcript levels of the transgene in selected transgenic $T_0$ plants. The $T_0$ and $T_1$ transgenic plants were challenged with eight Chinese *Pt* races, including FHJR, PHRT, PHTT, PHQS, FHJL, HCJR, THDB, and a mixture of naturally prevalent *Pt* races collected in 2021 from the field in Weifang, Shandong, China (36°26′04.0″N, 119°26′42.6″E). The number of integrated *Lr47* transgenes in each transgenic line was estimated based on the segregation ratio of $T_1$ transgenic plants and the results of a TaqMan copy number assay[30].

## qRT-PCR analysis

At the two-leaf stage, plants from the introgression line Kern *Lr47* were mock or *Pt* inoculated in two independent growth chambers under the same temperature and photoperiod (24 °C day/22 °C night and 16 h light/8 h dark). Leaf samples from different plants were collected immediately before inoculation (0 h) and at 1, 2, 4, and 6 dpi. Total RNAs were isolated using the Spectrum Plant Total RNA Kit (MilliporeSigma, MA, USA) and purified using the Direct-zol RNA MiniPrep Plus Kit (Zymo Research, CA, USA). qRT-PCR reactions were carried out on an ABI QuantStudio 5 Real-Time PCR System (Applied Biosystems, CA, USA) using Fast SYBR GREEN Master Mix. qRT-PCR primer pair *Lr47qPCRF2R3* (Supplementary Data 3) was used to evaluate the transcript levels of *Lr47*. Transcript levels were determined in at least seven biological replicates and presented as fold-*ACTIN* levels[58].

## Phylogenetic analysis

Sequences of Lr47 protein homologs were obtained from the National Center for Biotechnology Information (NCBI) database (https://www.ncbi.nlm.nih.gov) and the published reference genomes of wheat and its wild relatives (https://wheat.pw.usda.gov/blast/). To explore the origin of the introgressed segment containing *Lr47*, we performed RNA-seq and variant calling for seven *Ae. speltoides* genotypes, including T2140002, Y162, PI 542258, PI 542251, PI 560751, PI 542260, and PI 542270 (accession number PRJCA016987). Moreover, SNPs of four *Ae. speltoides* Tausch var. *speltoides* accessions (KU-2208A, KU-14601, KU-14605, and KU-12963a) and three *Ae. speltoides* Tausch var. *ligustica* accessions (KU-2236, KU-7716, and KU-7848) were obtained from RNA-seq data[12]. Multiple sequence alignment was conducted using the MUSCLE method implemented in MEGA software v7.0. The phylogenetic tree was generated using the pairwise deletion method (bootstrap values based on 1000 iterations) and visualized using the Interactive Tree Of Life (iTOL) v5.0 (https://itol.embl.de/).

## Reduction of *Ae. speltoides* chromosome segment

The introgression line Kern *Lr47* was crossed with CS*ph1b*[59] to induce homoeologous recombination and reduce the size of the *Lr47* introgressed chromosome segment 7S#1S. Markers *Xwgc2111* and *Xwgc2049* were used to validate the presence of the *ph1b* mutation[60]. The resulting $F_1$ plants from the cross Kern *Lr47* × CS*ph1b* were self-pollinated. The derived $F_2$ plants, which were heterozygous for the introgressed segment 7S#1S and homozygous for *ph1b*, were self-crossed to generate $F_3$ families. Using 7A/7S genome-specific markers, we identified recombination events within the introgressed alien chromosome segment and tested these plants for leaf rust resistance. The obtained recombinant plants heterozygous for *Lr47* and homozygous for *ph1b* (residual heterozygous lines) were self-crossed to generate $F_4$ progeny for another round of screening of new recombinants. The identified critical recombinants carrying *Lr47* were crossed and backcrossed to the Chinese hexaploid wheat variety YM21. Finally, plants homozygous for the shortened alien segments were selected

using molecular markers and evaluated for resistance to multiple *Pt* races.

## Cytogenetic assays

GISH and FISH were conducted at the Triticeae Research Institute, Sichuan Agricultural University[61]. The FISH probes oligo-pSc119.2, oligo-pTa535, and oligo-pTa713 were suitable for the identification of chromosomes of common wheat and *Aegilops* species[62]. Oligo-pSc119.2 preferentially paints tandem repeats on B- (or S-) genome chromosomes, and oligo-pTa535 hybridizes well to wheat A- and D-genome chromosomes[63]. A combination of oligo-pSc119.2 and oligo-pTa535 was used to distinguish the 42 wheat chromosomes[64]. Probe oligo-pTa713 generates additional diagnostic signals in the centromeric region of chromosome 7A[65]. The probes were labeled with either FAM or TAMRA and synthesized by TsingKe Biological Technology Co., Ltd. (Chengdu, China).

## Evaluation of agronomic and quality traits

Homozygous BC$_3$F$_3$ plants from the introgression line YM21-*Lr47*-1 and its sister line lacking the reduced alien chromosome introgression were grown in a greenhouse (22–30 °C day and 16–24 °C night with a 16 h photoperiod) and in a controlled walk-in growth chamber (24 °C day/22 °C night with a 16 h photoperiod). For the greenhouse and growth chamber experiments, a single plant was grown per pot (~3.8 L) and agronomic traits were measured for each plant. Tillers number (TN) was counted at the time of physiological maturity. Plant height (PH) was calculated by measuring the height of the main tiller of each plant from the ground level to the tip of spike excluding awns. Spike length (SL), spikelet number per spike (SNS), and grain number per spike (GNS) were measured as the mean of the main spikes (2–3 main spikes per plant). Grain yield per plant (GYP), thousand-seed weight (TSW), grain length (GL), and grain width (GW) were automatically calculated in the laboratory using a crop scanning test system (Wanshen SC-G, Hangzhou, China)[66]. At the maturity stage, the penultimate internodes were cut into segments of equal length (~10 cm), dried at 32 °C for 10 days, and then used for measurement. The shearing force (SF) of stems was measured using a universal testing machine (Instron 5848 Microtester, Instron, USA)[67]. Grain moisture content (GMC), grain protein content (GPC), flour water absorption (FWA), and flour ash (FA) were estimated using near-infrared reflectance spectroscopy (Model SpectraStar 2600 XT-R, Unity Scientific, USA). Grain hardness (GH) was determined using a TA-XT-plus Texture Analyzer (Stable Micro Systems, United Kingdom). Flour yield (FY) was expressed as grams of flour per 100 g of grain (conditioned to 15% moisture content, AACC 26-10). The field experiment was conducted during the 2022–2023 growing season at Peking University Institute of Advanced Agricultural Sciences, Weifang, China (36°26'04.0"N, 119°26'42.6"E). The introgression line YM21-*Lr47*-1 and its sister line were planted in 10 rows, each one meter long. The significance of differences in agronomic and quality traits between YM21-*Lr47*-1 and its sister line was estimated using two-sided unpaired *t* test.

## Yeast two-hybrid assay

The coding regions of the full-length Lr47 protein (CDS, amino acids 1–928), CC domain (amino acids 1–172), NB domain (amino acids 173–608), and LRR domain (amino acids 609–928) were cloned into the yeast two-hybrid system vectors pGBKT7 (bait) and pGADT7 (prey). The recombinant vectors were co-transformed into yeast strain AH109 using the lithium acetate and polyethylene glycol 3350 method. Co-transformed yeast was spotted onto synthetic dropout medium lacking leucine and tryptophan (SD-Leu-Trp) for selection of transformed colonies and then leucine, tryptophan, histidine, and adenine (SD-Leu-Trp-His-Ade) to detect interactions.

## *In planta* expression of GFP-fused Lr47 protein

The coding regions of the full-length Lr47 protein (amino acids 1–928) and CC domain (amino acids 1–172) were cloned into vectors pJIM19-GFP and pJIT163-Ubi-GFP[68,69]. The recombinant constructs were expressed in *N. benthamiana* leaves using transformed *A. tumefaciens* strain GV3101. Wheat protoplasts were isolated from the common wheat variety Fielder and transformed using the polyethylene glycol (PEG)-mediated method[70]. Fluorescence was checked using a confocal microscope (A1 HD25 Nikon, Tokyo, Japan). Total proteins were extracted from *A. tumefaciens*-transformed tobacco leaves. Protein samples were separated by sodium dodecyl sulfate-polyacrylamide gel electrophoresis (SDS-PAGE) and then transferred onto polyvinylidene difluoride (PVDF) membranes. Immunoblots were performed using an anti-GFP antibody. The primary antibody used for identification of GFP and GFP fusion proteins was Anti-GFP (Abcam, Cambridge, UK, Catalog No. ab290) at a dilution of 1:2500. Following that, the secondary antibody used to detect the primary antibody was Goat anti-Rabbit IgG-HRP (Abmart, Shanghai, China, Catalog No. M21002S) at a dilution of 1:8000.

For the cell death induction assay, the coding regions of the CC domain of a previously identified stem rust resistance protein Sr13[30] and a mammalian cell death inducer BAX[29] were cloned into pJIM19-GFP as positive controls. Highly concentrated (OD = 1.0) *A. tumefaciens* carrying Lr47_CDS-GFP, GFP-Lr47_CDS, Lr47_CC-GFP, GFP-Lr47_CC, Lr47_NB-GFP, GFP-Lr47_NB, Lr47_LRR-GFP, GFP-Lr47_LRR, Sr13_CC-GFP, BAX, and GFP were infiltrated into *N. benthamiana* leaves. Necrosis induced by BAX was observed at 48 h post inoculation (hpi), while yellowing caused by Sr13_CC-GFP was detected at 120 hpi.

## Reporting summary

Further information on research design is available in the Nature Portfolio Reporting Summary linked to this article.

## Data availability

Data supporting the findings of this work are available within the paper and its Supplementary Information files. All the raw sequencing data for this project are archived at the National Genomics Data Center under BioProject accession number PRJCA016987. The transcriptome assembly was deposited in Figshare [https://doi.org/10.6084/m9.figshare.23937879]. The sequence of the *Lr47* gene was deposited in NCBI Genbank under accession number OQ919262. Source data are provided with this paper.

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

## Acknowledgements

The authors thank Dr. Jan Dvorak (University of California, Davis, USA) for the initial *Aegilops speltoides* recombinant line; Dr. Bao Liu and Dr. Zhibin Zhang (Northeast Normal University, China) for genomic sequence of *Ae. speltoides* TS01 before release; and Dr. Xiuying Kong (Chinese Academy of Agricultural Sciences, China) and Dr. Lianquan Zhang (Sichuan Agricultural University, China) for *Ae. speltoides* accessions. Work at S.C. laboratory was supported by the Young Taishan Scholars Program of Shandong Province, the National Key Research and Development Program of China (2022YFD1201300), the Key R&D Program of Shandong Province (ZR202211070163), and the Provincial Natural Science Foundation of Shandong (ZR2021ZD30 and ZR2021MC056). Work at X.W. laboratory was supported by Provincial Natural Science Foundation of Hebei (C2022204010 and C2021204008). Work at J.D. laboratory was supported by the Howard Hughes Medical Institute and by the Agriculture and Food Research Initiative Competitive Grant 2022-68013-36439 (WheatCAP) from the USDA National Institute of Food and Agriculture (NIFA).

## Author contributions

H.L., L.H., and S.Z. performed most of the experimental work; M.H. and H.C. conducted cytogenetic assays; R.S., Y.L., and X.H. contributed to subcellular localization; S.P., T.S., G.W., and J.L. contributed to primer development; W.Z. validated the diagnostic marker in *Ae. speltoides*; J.-Y.G. contributed to EMS population; H.M. and C.L. evaluated quality traits; B.S. contributed to AlphaFold prediction; Z.L. contributed to phenotyping; X.W.D. provided scientific support; S.C., X.W., and H.L. analyzed the data and wrote the first version of the manuscript. S.C., X.W., and J.D. conceived and supervised the project. All authors reviewed and revised the manuscript.

## Competing interests

S.C., H.L., L.H., X.W., R.S., and Y.L. are inventors on a Chinese provisional patent application (China patent filing No.202211247709.2) relating to the use of the *Lr47* gene in wheat breeding programs. The remaining authors declare no competing interests.
