## [Peer Review File · Nature Communications]

Cloning of the wheat leaf rust resistance gene Lr47
introgressed from *Aegilops speltoides*Reviewers' Comments:

Reviewer #1:

Remarks to the Author:

In this study the sequence corresponding to the leaf rust resistance gene Lr47 is identified. First, the effectivity of the chromosome translocation from *Aegilops speltoides* carrying Lr47 was tested against 23 Chinese pathotypes of *Puccinia triticina*, the fungus causing wheat leaf rust. Then Lr47 NILs and its recurrent parents were genotyped to define the position and length of the translocation in chromosome 7AS. The wheat mutation ph1b was introduced into one of these NILs and used to promote recombination between wheat chromosome 7A and the *Ae. speltoides* 7S segment thus reducing the length of the 7S fragment carrying Lr47 from 150 Mb to 2.5 Mb. In addition, EMS mutagenesis of the Lr47 translocation and screening with the Pt race THDB identified ten independent susceptible mutant families. The authors cloned Lr47 by comparing RNA-seq data from these mutants against transcripts of genes in the Lr47 interval. Through this comparison, a single gene was identified with canonical EMS-type transition mutations in all the mutants. The gene was further validated by engineering CRISPR-Cas9 loss-of-function and generating stable transgenics in cultivar Fielder. Finally, a BC3F3 recombinant line carrying the reduced Lr47 fragment was studied for agronomic and quality characteristics with its sister line lacking the translocation. Under disease-free conditions no differences were observed except for shorter spikes in the line carrying the translocation. In conclusion, the data fully support that the correct gene has been identified, and that the introgressed segment has been shortened. This makes Lr47 an attractive target for conventional breeding and for incorporation into polygene GM stacks for control of wheat leaf rust.

The manuscript is well written, shows originality and the statements made are generally backed by good quality data. The text can be easily read, and shows coherence with the figures and tables. The subject is likely to garner substantial interest in the wheat breeding and disease resistance community as the number of cloned leaf rust resistant genes is small. Given the huge importance of leaf rust as the main biotic constraint to wheat cultivation, this paper is likely to contribute to the development of strategies to improve global food security.

Major comment:

One important qualifier should be made regarding the attribution of broad-spectrum resistance to Lr47 (e.g. title, abstract, line 30, and lines 115-116). Whereas the authors clearly demonstrate that the translocation carrying the Lr47 gene displays broad-spectrum resistance to multiple Chinese isolates of the pathogen, there is, however, no conclusive evidence that the isolated Lr47 gene mirrors the same level of resistance found in the 7A/7S original translocation. It cannot be formally excluded that there are other genes in the translocation, in addition to Sr47, that contribute to the resistance, similar to e.g. the case of the original Sr32 introgression in which another Sr gene was later found to contribute to resistance (Mago et al., 2013; <https://pubmed.ncbi.nlm.nih.gov/23989672/>). To substantiate the broad-spectrum claim, the authors would have to test the mutants and/or the transgenics against the same range of wheat leaf rust isolates used on the translocation line. In the work presented here the transgenic plants carrying Lr47 were tested only against one single strain of *Puccinia triticina*. If the authors are not in a position to complete further pathology analysis of the mutant/and or transgenic lines in a timely manner, then we suggest that the broad-spectrum claim is removed from the title and countered with the alternate hypothesis in the text, i.e. that the translocation line may contain other resistance determinants.

Minor comments

Line 35: explaining the meaning of the CC abbreviation in the abstract would facilitate the understanding for those readers outside of the field.

Line 49: the reference #5 does not relate with the annual global wheat yield losses due to leaf rust.

Line 96: tautology produced with the word cultivar and the abbreviation cv.

Line 101: consider explaining the meaning of the dpi abbreviation to facilitate the reading.

Line 110: the phrase "...were challenged with the other 23 Pt pathotypes..." is confusing as the 23 Pt pathotypes are not mentioned before.

Line 112: explaining the meaning of the IT abbreviation would facilitate the reading.

Line 114: and Fig. 1d: from the text the reader is led to expect finding a picture of the infection reaction in the Yecora Rojo recurrent parent, but that parent is not shown in the figure.

Line 122: it would be interesting to know the origin of the three Ae. speltoides accessions used.

Line 159: To back up their claim that most R genes are NLRs, the authors cite a few studies (predominantly their own) that have cloned resistance genes that turned out to be NLRs. The authors would do better to cite a review that has gone to the length of surveying the several hundred resistance genes that have been cloned since 1992 and from which such an NLR claim can be substantiated.

Line 161: the sudden mention of the EMS mutant line m118 with no reference to where it came from is surprising – a bit like a magician taking a rabbit out of their hat. It should be clear from the text whether this is a mutant that was generated as part of the present study, or one that was obtained from a previous study. If the mutant is from the present study, then this gives rise to an anachronism in the narrative, as the mutagenesis is introduced later. This needs to be fixed.

Line 205: based on the model presented for the secondary structure of the Sr47 protein, the number of leucine-rich repeat units forming the typical α/β horseshoe fold could be specified.

Line 268: Ae. Longissima is misspelled.

Line 355: an explanation of the abbreviations in the text for SNS, PH, and TN could facilitate the reading.

Line 346-348: if the data is available (and only if), it would be interesting to have the comparison between sister lines without and with the translocation, and with the full and short fragment versions. This could provide a very nice exemplar of the effect of linkage drag.

Line 372-379: the motivation for the experiments described in this paragraph is somewhat unclear, and the results are also somewhat preliminary. If the manuscript in its present form is found to be too long, then this could be removed without affecting the quality of the manuscript.

Line 389: 12 independent lines are mentioned in this section but in the results section (line 173) and in the methods (line 534) only 10 mutants are mentioned.

Line 87, 171-172, 388-393 and 529: the authors used mutational transcriptomics to clone Lr47 and name their method "EMS mutagenesis and transcript assembly (EMTA)". Instead of billing this as a novel enabling technology, the authors would do justice here to cite previous papers which have used similar methods to clone resistance genes in wheat, e.g. MutRNASeq for cloning Sr62 (Yu et al., 2022, Nature Communications 13, Article number: 1607) and MutIsoSeq for cloning Lr9 (Wang et al., 2023, Nature Genetics, doi: 10.1038/s41588-023-01401-2). In particular, Figure 3 is very similar to the MutRNAseq Figure in Yu et al., 2022.

Line 454-457: in the text the Sr13 is not mentioned, while it is used as an example in the supplemental Fig. 23.

Line 589: "Multiple" is misspelled.

Line 904-912: please explain meaning of Express, RSI5 and UC1041 in the legend.

Line 914-923: please explain the meaning of the lower- and upper-case letters that refer to NLRs in Fig. 2e.

Figure 6: scatter box plots would represent more appropriately the variability of the data.

Figure 7: this figure and the corresponding paragraph in the text present somewhat preliminary results. If the manuscript needs to be shortened, then this section could be omitted without affecting the quality of the manuscript. Scatter box plots would represent more appropriately the variability of the data in Figure panel a.

Supplementary Fig. 4: please remove white triangles in the figure corresponding to m1541 to facilitate the interpretation of the graph.

Supplementary Fig. 9: show the error bars but number of samples (n) is not shown, please detail n.

Supplementary Fig. 13 and 14: refers MAGA 7 instead of MEGA 7, please correct the wrong name.

Supplementary Table 3: "responses to Lr47" is ambiguous, please define what germplasm was used in the table.

Reviewer #2:

Remarks to the Author:

Leaf rust is a severe disease of wheat all over the world. Development of resistant varieties is an effective way to control this disease, so identification of the broad-spectrum Lr resistance gene shows great value. Wild species are valuable resources for mining resistance genes and many resistance genes have been transferred to wheat by chromosome engineering. However, due to the low recombination of chromosomes from wild species with that from wheat, it is not easy to clone such kind of genes. In addition, linkage drag caused by large segments introgression usually limited utilization of wild germplasm in breeding. Exactly, few resistance genes from wild species have been successfully used in wheat. In this study, the broad-spectrum resistance gene Lr47 was cloned by fine mapping, mutants creation, gene knockout and stable transformation. The manuscript provided strong evidence to support the conclusion that NLR2 is the Lr47 gene. Moreover, the germplasm with small introgression segment was also identified for wheat breeding. Therefore, this research provided Lr47 gene for resistance improvement by genetic engineering and germplasm for resistance improvement by traditional backcrossing. I think this research may be of particular interest to the readers of Nature Communication and I recommend for its publication after revision.

Some other comments as follows,

1. In Figure 1, the average infection areas were compared between plants with Lr47 and without Lr47 to show the difference of pathogen development. It was found that no cell death was activated in tobacco by CC-domain of Lr47. Could the author provide images to show infected cells of plants with Lr47 and without Lr47 to determine whether auto-fluorescence produced due to cell death activated in infected cells of wheat.

2. In Figure 2a, it was not appropriate to using 'chr. 7A' to represent the chromosome introgression with alien chromosome segment.

3. In Figure 4, T1CS401-2 (1 copy) displayed stronger HR compared to T1CS401-1 (3 copies). Similarly, in Supplemental Figure 11, T1CS401-7 (1 copy) displayed stronger HR compared to T1CS401-10 (5 copies). Why did such difference occur? Is there difference in gene expression level among different lines with varied insertion copies?
4. In Figure 5b, the author indicated the 7A chromosome (painted in red) with introgression segment of 7S (painted in green) marked with yellow arrowheads. Another pair of chromosomes also showed both red and green signals, and one of which was near closely to the arrowed chromosome. The author should explain why this pair of chromosomes produced green signals?
5. In addition, ph1b could induce recombination between 7S and 7A, it could also induce recombination between other homoeologous chromosomes of wheat. It was obvious to find large segment of translocation between D and A chromosomes in the Figure 5d. The author should analyze the structure variations of all the involved chromosomes including wheat chromosomes.
6. In Supplementary Fig. 10, all of the transgenic plants showed high resistance to Pt. Why did some of the leaves show strong HR while some leaves show no HR as observed in Kern Lr47?
7. In Supplemental Figure 21, the scale bar was not clear.
8. In line 361, 'indicating that Lr47 is not induced by the presence of the Pt pathogen'. I suggested the author provided the leaves of Kern Lr47 with Pt inoculation and without Pt inoculation to compare the HR reaction before and after Pt infection.
9. A total of 23 Pt has been tested for their virulence to Lr47. Which type of the races are the currently virulent races threatening wheat production? I suggested the author supply the data about Pt races used in the manuscript to help breeders evaluating the value of Lr47 in current wheat breeding.
10. Lr47 was characterized to be an NLR as a lot of cloned resistance genes from wheat and its relatives. In this study, there is little resistance mechanism revealed about Lr47. I suggest it's better to provide some special mechanism about the broad-spectrum resistance of Lr47.

Reviewer #3:

Remarks to the Author:

Wheat leaf rust is one of the most severe foliar diseases in wheat production worldwide. Lr47 derived from a wild relative of wheat, *Aegilops speltoides*, provides broad-spectrum resistance. Lr47 was cloned by a combination of map-based cloning and EMS mutagenesis and transcript assembly (EMTA) approaches. Lr47 encodes a CC NLR protein, and was confirmed by mutation and transformation studies. From the practical point of view, the authors developed lines with smaller alien introgression, which did not have linkage drag. Markers were developed to be used in marker-assisted selection. In general, the manuscript was well written. Even though the gene cloned has a structure of an NLR-type, the findings in this paper have advanced our knowledge from both the fundamental and applied aspects. Hopefully the line with small introgression will be deployed to provide additional source for resistance breeding!

Here are additional points for revision.

Either in the "Abstract" or "Introduction" or both, it will be good to mention that Lr47 is a seedling / all-stage resistance gene, for the interest of readers.

It will be good if the authors can add in the "Introduction" whether any virulence to Lr47 has been found anywhere in the world.

L70: Better use "homoeologous1".

L244-247: What are the reasons resulting in the deviation from the 3:1 segregation ratio? What might be the reasons for preferential transmission, segregation distortion?

L268: Should be "longissima"; "83.29-92.78%".

L285: Change "that" to "which".

L355: Spell out "SNS, PH, TN" because they were not mentioned before.

L366: Use "Nicotiana" here and change to "N." in L649.

L433: To follow the current taxonomy, better use "Thinopyrum ponticum" instead of "Ag. elongatum".

L473: Change to "Lr34/Sr57/Yr18 and Lr67/Sr55/Yr46".

L510: How many seeds were treated?

L607: The two references "75 and 76" cited did not have method for GISH, better add or change.

L610: oligo-pTa535 also hybridized to A-genome chromosomes in addition to D-genome ones.

L612-613: Need to explain better, e.g. "probe oligo-pTa713 generates additional diagnostic signals..." However, in the images (Fig. 5e, f, g), yellow signals of pTa-713 were almost invisible. In general, the FISH signals were not good, too weak. Fig. 5e and f are acceptable, but not Fig. 5d and g, which need a better cell.

Fig. 2c: It will be good to add the diagram of L8 here for comparison.

Fig. 4d and Supplementary Fig. 11: What's the reason for the susceptible (having 1 copy of transgene) plants in T1CS401-2 and T1CS401-7?

Fig. 6s: The spike length difference is not obvious. I am wondering whether it is better not to include this figure.

In the figure legend for Fig. 7e and Supplementary Fig. 22: It is better to add "No cell death was observed for".

Supplementary Fig. 19: No need to show (a) and (b) because they did not provide much information.

Supplementary Fig. 20: Not necessary as a figure. Suggest delete.

Supplementary Table 3: The avr/vir profiles are missing for the last four pathotypes.

Reviewer #4:

Remarks to the Author:

This study made tremendous efforts to clone a leaf rust gene, Lr47, in wheat. Lr47 represents one of a few effective leaf rust resistance genes in wheat and the chromosomal segment harboring Lr47 was transferred from *Aegilops speltoides*. Using two mapping wheat populations, the gene was mapped to a certain region including multiple NLR genes. Transcriptomes of EMS mutants were generated and assembled to obtain transcript sequences. Combination of mapping data, the reference genome, assembled transcripts, and potential EMS mutations led to a candidate gene, which was verified by sequence confirmation of EMS mutants, CRISPR knockouts through virus-mediated mutagenesis, and stable ectopic expression. The linkage drag introduced by the translocation segment was reduced by shortening the translocation segment through homoeologous recombination in the mutant of the ph1 gene. In addition, a diagnostic molecular marker was developed to identify the potential Lr47 donor, which can be also used to facilitate the deployment of the gene in breeding programs. Overall, the

manuscript was well written and provided solid data for genetic mapping and candidate validation. I do not have major concerns regarding to the experimental design or data interpretation. I have the following minor comments for authors' references.

1. The ethyl methanesulfonate (EMS) mutagenesis and transcript assembly (EMTA) approach was highlighted in the manuscript. In my opinion, this is a fairly regular and straightforward approach. It is not necessary to be highlighted.
2. L75: when chromosomal coordinates are described, it would be informative if the reference genome (species and genome version) is specified.
3. L129: I might miss something, But I was confused when I read the paragraph from L118-130. How was the following conclusion drawn. "... Thus, the *Ae. speltoides* segment in the three Lr47 NILs ranges from 150.1 Mb to 151.8 Mb". Sorry, I just did not see the logic.
4. L161: When m118 was first mentioned, the EMS experiment has not described. It would be clearer to mention that the EMS experiment was performed and one of the mutants was used to build a mapping population.
5. L202: To be clear, "the mutations" can be replaced by "the EMS mutations".
6. L376: It would be helpful to briefly describe the purpose of BAX and Sr13 in "expressing BAX and the CC domain of Sr13".
7. The abbreviation of the first appearance of a term should be spelled out and followed by its abbreviation in parentheses, such as Genomic In Situ Hybridization (GISH). I saw the full name in the Methods section, which is located behind the Results section, it is appropriate to use the abbreviation thereafter.

We have incorporated all the suggestions/comments provided by the reviewers. The reviewer's comments are listed below and marked in *Italics*, response to the reviewer's comments are marked below in black, while the edited/revised/new text from the paper is marked below in blue.

Reviewer #1:

In this study the sequence corresponding to the leaf rust resistance gene Lr47 is identified. First, the effectivity of the chromosome translocation from Aegilops speltoides carrying Lr47 was tested against 23 Chinese pathotypes of Puccinia triticina, the fungus causing wheat leaf rust. Then Lr47 NILs and its recurrent parents were genotyped to define the position and length of the translocation in chromosome 7AS. The wheat mutation ph1b was introduced into one of these NILs and used to promote recombination between wheat chromosome 7A and the Ae. speltoides 7S segment thus reducing the length of the 7S fragment carrying Lr47 from 150 Mb to 2.5 Mb. In addition, EMS mutagenesis of the Lr47 translocation and screening with the Pt race THDB identified ten independent susceptible mutant families. The authors cloned Lr47 by comparing RNA-seq data from these mutants against transcripts of genes in the Lr47 interval. Through this comparison, a single gene was identified with canonical EMS-type transition mutations in all the mutants. The gene was further validated by engineering CRISPR-Cas9 loss-of-function and generating stable transgenics in cultivar Fielder. Finally, a BC3F3 recombinant line carrying the reduced Lr47 fragment was studied for agronomic and quality characteristics with its sister line lacking the translocation. Under disease-free conditions no differences were observed except for shorter spikes in the line carrying the translocation. In conclusion, the data fully support that the correct gene has been identified, and that the introgressed segment has been shortened. This makes Lr47 an attractive target for conventional breeding and for incorporation into polygene GM stacks for control of wheat leaf rust.

The manuscript is well written, shows originality and the statements made are generally backed by good quality data. The text can be easily read, and shows coherence with the figures and tables. The subject is likely to garner substantial interest in the wheat breeding and disease resistance community as the number of cloned leaf rust resistant genes is small. Given the huge importance of leaf rust as the main biotic constraint to wheat cultivation, this paper is likely to contribute to the development of strategies to improve global food security.

Author's Answer: Thank you for taking the time to evaluate our manuscript. We greatly appreciate your positive feedback and valuable insights into the study.

Major comment:

One important qualifier should be made regarding the attribution of broad-spectrum resistance to Lr47 (e.g. title, abstract, line 30, and lines 115-116). Whereas the authors clearly demonstrate that the translocation carrying the Lr47 gene displays broad-spectrum resistance to multiple Chinese isolates of the pathogen, there is, however, no conclusive evidence that the isolated Lr47 gene mirrors the same level of resistance found in the 7A/7S original translocation. It cannot be formally excluded that there are other genes in the translocation, in addition to Lr47, that contribute to the resistance, similar to e.g. the case of the original Sr32 introgression in which another Sr gene was later found to contribute to resistance (Mago et al., 2013; <https://pubmed.ncbi.nlm.nih.gov/23989672/>). To substantiate the broad-spectrum claim, the authors would have to test the mutants and/or the transgenics against the same range of wheat leaf rust isolates used on the translocation line. In the work presented here the transgenic plants carrying Lr47 were tested only against one single strain of Puccinia triticina. If the authors are not in a position to complete further pathology analysis of the mutant/and or transgenic lines in a timely manner, then we suggest that the broad-spectrum claim is removed from the

title and countered with the alternate hypothesis in the text, i.e. that the translocation line may contain other resistance determinants.

Answer: We thank the reviewer for the valuable comment. This is a very valid request. We screened eight *Pt* races in eight independent growth chambers and found that the Fielder control was susceptible to seven *Pt* races, including FHJR, PHRT, PHTT, PHQS, FHJL, HCJR, and a mixture of naturally prevalent *Pt* races collected from the field in 2021 (36°26'04.0"N, 119°26'42.6"E). In contrast, the transgenic plants carrying the *CNL2* transgene displayed resistant reactions (similar to the introgression line Kern *Lr47*) to all the *Pt* races tested, as shown in new Figure 4d. These results confirmed that transgenic plants carrying the *CNL2* transgene is broadly effective against all leaf rust races tested. However, although no virulence was found to *Lr47* so far, it has also not been widely used in agriculture. Thus, we agree with the reviewer here and removed the “broad-spectrum” claim from the title (being cautious).

For the EMS mutants, the recurrent parent Kern was reported to possess unknown *Pt* resistance gene(s) and was resistant to several leaf rust races (Chicaiza et al. 2006; <https://escholarship.org/uc/item/6tn2f0dg>). In our study, we screened five *Pt* races and found that Kern was only susceptible to races THDB and PHQS (as new Supplementary Fig. 5). Therefore, we screened EMS mutants using *Pt* race THDB. Then, we confirmed the susceptibility to races THDB and PHQS by evaluating M₃ seeds derived from the susceptible M₂ plants (Fig. 3a and Supplementary Fig. 6a). The obtained susceptible EMS mutants **were completely susceptible** when inoculated with *Pt* race THDB (Fig. 3a), indicating no other genes in the translocation conferring resistance to race THDB.

We added these results (or rephrased the sentences) in the “Results” section as follows:

Lines 168-170: “Since the recurrent parent Kern was susceptible to *Pt* races THDB and PHQS (Supplementary Fig. 5), these two races were subsequently used to identify susceptible EMS mutants of Kern *Lr47*.”

Lines 265-269: “Transgenic T₁ plants from the 11 selected transgenic events were challenged with seven *Pt* races, all of which are virulent on Fielder. All plants from the T₁ transgenic families T₁CS401-1 and T₁CS401-10, which were fixed for the transgene, exhibited a high level of resistance, and resistance in the transgenic families T₁CS401-2 and T₁CS401-7 perfectly co-segregated with the presence of the transgene (Fig. 4d and Supplementary Fig. 13).”. We also revised the other part of the manuscript accordingly.

Minor comments

Line 35: explaining the meaning of the CC abbreviation in the abstract would facilitate the understanding for those readers outside of the field.

Answer: This change was made as suggested.

Line 49: the reference #5 does not relate with the annual global wheat yield losses due to leaf rust.

Answer: We deleted the reference #5 and the sentence.

Line 96: tautology produced with the word cultivar and the abbreviation cv.

Answer: We deleted “cv.”

Line 101: consider explaining the meaning of the dpi abbreviation to facilitate the reading.

Answer: Corrected as suggested.

Line 110: the phrase "...were challenged with the other 23 Pt pathotypes..." is confusing as the 23 Pt pathotypes are not mentioned before.

Answer: The sentence was rephrased as follows (lines 114-116): "Seedlings of *Lr47* near-isogenic lines (NILs; Express *Lr47*, UC1041 *Lr47*, and RSI5 *Lr47*) and their recurrent parents (Express, UC1041, and RSI5) were challenged with 23 additional *Pt* pathotypes collected in China (Supplementary Table 1)."

Line 112: explaining the meaning of the IT abbreviation would facilitate the reading.

Answer: Corrected as suggested.

Line 114: and Fig. 1d: from the text the reader is lead to expect finding a picture of the infection reaction in the Yecora Rojo recurrent parent, but that parent is not shown in the figure.

Answer: We deleted "Yecora Rojo" from the text, since we didn't have good pictures for this recurrent parent and its NIL-*Lr47*.

Line 122: it would be interesting to know the origin of the three Ae. speltooides accessions used.

Answer: As shown in Supplementary Table 2, the origin of AE915 and AE1590 are unknown, and PI 554292 was collected in Turkey.

The sentence was rephrased as follows (lines 125-129): "we compared the single nucleotide polymorphisms (SNPs) identified in the RNA-seq of *Lr47* NILs (Kern *Lr47*, Yecora Rojo *Lr47*, and UC1041 *Lr47*) with those from another ten sequenced hexaploid wheat varieties and three *Ae. speltooides* accessions (AE915, AE1590, and PI 554292; Supplementary Table 2)."

Line 159: To back up their claim that most R genes are NLRs, the authors cite a few studies (predominantly their own) that have cloned resistance genes that turned out to be NLRs. The authors would do better to cite a review that has gone to the length of surveying the several hundred resistance genes that have been cloned since 1992 and from which such an NLR claim can be substantiated.

Answer: Corrected as suggested.

Line 161: the sudden mention of the EMS mutant line m118 with no reference to where it came from is surprising – a bit like a magician taking a rabbit out of their hat. It should be clear from the text whether this is a mutant that was generated as part of the present study, or one that was obtained from a previous study. If the mutant is from the present study, then this gives rise to an anachronism in the narrative, as the mutagenesis is introduced later. This needs to be fixed.

Answer: The EMS mutant line "m118" is a mutant generated in the present study. To resolve the problem, we moved this paragraph to the next section, and described the m118 x Kern *Lr47* (WT) mapping population after obtaining susceptible EMS mutant lines.

We rephrased the sentence as follows (lines 176-178): "To refine the mapping of *Lr47* in the Kern *Lr47* × *CSph1b* population, we crossed one of the obtained susceptible EMS mutant lines, m118, with the non-mutagenized resistant Kern *Lr47*, and generated an F₂ population consisting of 1,141 individuals...."

Line 205: based on the model presented for the secondary structure of the Lr47 protein, the number of leucine-rich repeat units forming the typical α/β horseshoe fold could be specified.

Answer: Agree. Corrected as suggested.

Line 268: *Ae. Longissima* is misspelled.

Answer: Corrected as suggested.

Line 355: an explanation of the abbreviations in the text for SNS, PH, and TN could facilitate the reading.

Answer: We deleted this sentence from the manuscript.

Line 346-348: if the data is available (and only if), it would be interesting to have the comparison between sister lines without and with the translocation, and with the full and short fragment versions. This could provide a very nice exemplar of the effect of linkage drag.

Answer: We didn't have the data from sister lines with and without the full fragment carrying Lr47. Agronomic and quality evaluation of common wheat near-isogenic lines carrying the full introgressed chromosomal fragment was reported previously by Brevis *et al.* (2008). In this study, we only have a comparison between sister lines with and without the shortened alien chromosome segment.

Line 372-379: the motivation for the experiments described in this paragraph is somewhat unclear, and the results are also somewhat preliminary. If the manuscript in its present form is found to be too long, then this could be removed without affecting the quality of the manuscript.

Answer: The manuscript is not too long, and meets the length requirement of Nature Communications. For CC NLRs, the CC domain is thought to play an important role in signaling. Previous studies reported that the CC domains of many NLR proteins (such as MLA10, Pm21, Sr33, and Sr50) are sufficient to cause cell death after transient expression in *N. benthamiana* leaves. However, there are other NLR proteins, including RPM1, RPS5, Rx, and Bs2, whose CC domains do not induce cell death. Self-interaction was detected for some NLR genes, such as the rice Resistance Gene Analog 5 (RGA5) and the barley powdery mildew resistance protein MLA10. In contrast, no self-interaction was detected for some other NLR genes. Thus, it is good to understand the similarities and differences in the way Lr47 operates. Therefore, we performed a functional analysis of Lr47 focusing on their ability to induce cell death and self-interact.

Line 389: 12 independent lines are mentioned in this section but in the results section (line 173) and in the methods (line 534) only 10 mutants are mentioned.

Answer: We apologize for the confusion. In fact, we got 10 susceptible EMS mutant lines and 2 BSMV-based editing mutants. To avoid ambiguity, we rephrased the sentence as follows (lines 409-412): “The susceptibility to *Pt* races of the mutants induced by EMS or BSMV-mediated gene editing and the resistance of the *CNL2* transgenic Fielder lines (Figs. 3 and 4) confirmed that this candidate gene is both necessary and sufficient to confer resistance to leaf rust,”

Line 87, 171-172, 388-393 and 529: the authors used mutational transcriptomics to clone Lr47 and name their method “EMS mutagenesis and transcript assembly (EMTA)”. Instead of billing this as a novel enabling technology, the authors would do justice here to cite previous papers which have used similar methods to clone resistance genes in wheat, e.g. *MutRNASeq* for cloning Sr62 (Yu *et al.*, 2022, *Nature Communications* 13, Article number: 1607) and *MutIsoSeq* for

cloning *Lr9* (Wang et al., 2023, *Nature Genetics*, doi: 10.1038/s41588-023-01401-2). In particular, Figure 3 is very similar to the MutRNAseq Figure in Yu et al., 2022.

Answer: We agree with the Reviewer here. We have deleted “EMS mutagenesis and transcript assembly (EMTA)” from the manuscript and cited the previous paper about cloning of the stem rust resistance gene *Sr62* (Yu et al., 2022).

However, compared with the previously reported MutRNASeq approach, there are differences between our method and the previous MutRNASeq method. In our Methods section (lines 552-558), we described the procedure of our method: “The modified MutRNASeq was performed as follows: RNA-seq reads from Kern *Lr47* were assembled *de novo* using Spades version 3.14.1⁵⁴. We performed BLASTN searches of the transcriptome database of Kern *Lr47* using sequences of typical NLR genes within the *Lr47* candidate regions from the reference genomes of CS and TS01⁵⁵ as queries, and obtained transcript contigs with similarity values greater than 85% to the homologs or paralogs within the interval that *Lr47* was mapped to. Next, the RNA-seq reads of the ten susceptible EMS mutants were mapped to these transcript contigs as reported previously²⁴.”

The key difference is that our method “does not require the generation of a physical map/reference sequence across the map interval (line416)”. Since the mapping candidate regions in TS01 and CS reference genomes have a cluster of NLR genes, we hypothesized that *Lr47* might be an NLR genes, and then we performed the following analysis -- as described in the Results section (lines190-196): “Assembly of the Kern *Lr47* sequenced reads yielded 146,715 high confidence transcript contigs (\geq 500 bp). We performed BLASTN searches of the Kern *Lr47* transcriptome database using the sequences of the candidate NLR genes in CS and TS01 as queries, and obtained 45 transcript contigs with a BLAST e-value = 0. We mapped the sequenced reads of ten susceptible mutants against the 45 selected Kern *Lr47* transcript contigs and found one contig named KN638873_g379_i12 with EMS-type (G/C-to-A/T) point mutations in all ten mutants (Supplementary Figs. 6b and 7a).” Thus, we would like to term this as “modified MutRNASeq” in the current version R1 (but not bill this as a novel technology).

Line 454-457: in the text the *Sr13* is not mentioned, while it is used as an example in the supplemental Fig. 23.

Answer: Thanks for pointing it out. We described *Sr13* in the Methods section (lines688-690): “For the cell death induction assay, the coding regions of the CC domain of a previously identified stem rust resistance protein *Sr13*³⁰ and a mammalian cell death inducer BAX²⁹ were cloned into pJIM19-GFP as positive controls.”

Line 589: “Multiple” is misspelled.

Answer: Corrected as suggested.

Line 904-912: please explain meaning of Express, RSI5 and UC1041 in the legend.

Answer: UC1041, Express, and RSI5 are recurrent parents. We have revised the legend to read as follows: “Infection types of *Lr47* NILs and their recurrent parents (UC1041, Express, and RSI5) in response to selected *Pt* races FHJL, PHQS, FHJR, THDB, PHRT, PHTT, THTT, HCJR, and FHHM.”

Line 914-923: please explain the meaning of the lower- and upper-case letters that refer to NLRs in Fig. 2e.

Answer: Corrected as suggested. We added the following sentence to the legend of Fig. 2e: “NLR genes are represented by yellow arrows and upper-case letters (*NLR*) and pseudogenes by lower-case letters (*nlr*).”

Figure 6: scatter box plots would represent more appropriately the variability of the data.

Answer: Corrected as suggested.

Figure 7: this figure and the corresponding paragraph in the text present somewhat preliminary results. If the manuscript needs to be shortened, then this section could be omitted without affecting the quality of the manuscript. Scatter box plots would represent more appropriately the variability of the data in Figure panel a.

Answer: Thanks for pointing it out. Corrected as suggested.

Supplementary Fig. 4: please remove white triangles in the figure corresponding to m1541 to facilitate the interpretation of the graph.

Answer: Corrected as suggested.

Supplementary Fig. 9: show the error bars but number of samples (n) is not shown, please detail n.

Answer: Corrected as suggested.

Supplementary Fig. 13 and 14: refers MAGA 7 instead of MEGA 7, please correct the wrong name.

Answer: Corrected as suggested.

Supplementary Table 3: “responses to Lr47” is ambiguous, please define what germplasm was used in the table.

Answer: We apologize for not giving a clear description of the legend. We rephrased the Title of Supplementary Table 3 (now Supplementary Table 1) as follows: “**Supplementary Table 1. Avirulence/virulence formulae of *Pt* races used in this study. To determine the resistance profiles of *Lr47*, seedlings of *Lr47* NILs (Express *Lr47*, UC1041 *Lr47*, and RSI5 *Lr47*) and their recurrent parents (Express, UC1041, and RSI5) were challenged with these *Pt* races. *Lr47* is underlined.**”

Reviewer #2:

*Leaf rust is a severe disease of wheat all over the world. Development of resistant varieties is an effective way to control this disease, so identification of the broad-spectrum Lr resistance gene shows great value. Wild species are valuable resources for mining resistance genes and many resistance genes have been transferred to wheat by chromosome engineering. However, due to the low recombination of chromosomes from wild species with that from wheat, it is not easy to clone such kind of genes. In addition, linkage drag caused by large segments introgression usually limited utilization of wild germplasm in breeding. Exactly, few resistance genes from wild species have been successfully used in wheat. In this study, the broad-spectrum resistance gene *Lr47* was cloned by fine mapping, mutants creation, gene knockout and stable transformation. The manuscript provided strong evidence to support the conclusion that NLR2 is the *Lr47* gene. Moreover, the germplasm with small introgression segment was also identified for wheat breeding. Therefore, this research provided *Lr47* gene for resistance improvement by genetic engineering and germplasm for resistance improvement by traditional backcrossing. I think this research may be of particular interest to the readers of Nature Communication and I recommend for its publication after revision. Some other comments as follows,*

Answer: Thank you for your positive review of our manuscript.

1. In Figure 1, the average infection areas were compared between plants with *Lr47* and without *Lr47* to show the difference of pathogen development. It was found that no cell death was activated in tobacco by CC-domain of *Lr47*. Could the author provide images to show infected cells of plants with *Lr47* and without *Lr47* to determine whether auto-fluorescence produced due to cell death activated in infected cells of wheat.

Answer: As suggested by the reviewer, we have provided histological images that show cells infected with *Pt* race THDB at 2 and 4 days post-inoculation (dpi) in Kern *Lr47* (as new Supplementary Figure 1). The observed accumulation of phenolic auto-fluorogens indicates the activation of cell death at 2 dpi, which then increased significantly at 4 dpi. No substantial cell death was observed in the susceptible Kern control. Based on these observations, we speculate that there is a certain mechanism required for the activation of cell death controlled by *Lr47* in wheat plants, but not tobacco leaves, that is possibly associated with avirulent effectors from the leaf rust fungus. It is a common phenomenon that the CC domain of NLR proteins cannot induce cell death in tobacco, as demonstrated in previous studies on *Sr35*, *RPM1*, *RPS5*, *Rx*, and *Bs2*. We have added these results and revised the manuscript accordingly.

We added the following sentences in the Results section (lines 110-113): “Microscopic examination of leaf samples from Kern *Lr47* also revealed the accumulation of phenolic auto-fluorogens at each infection site, starting from 2 dpi, and showing a substantial increase at 4 dpi (Supplementary Fig. 1).”. We also added the description in the Methods section (lines 520-524): “To estimate the growth of the *Pt* pathogen and cell death triggered by *Lr47* in NILs, leaf segments (~6 cm long) from Kern *Lr47* and its recurrent parent Kern inoculated with race THDB were sampled at 2, 4, 6, and 8 dpi. Following bleaching with a 1:1 mixture of ethanol and glacial acetic acid, and subsequent transparency using chloral hydrate, the direct observation of phenolic auto-fluorogens' accumulation at each infection site is possible under epi-fluorescence microscopy.”

2. In Figure 2a, it was not appropriate to using ‘chr. 7A’ to represent the chromosome introgression with alien chromosome segment.

Answer: Thanks for pointing it out. The ‘chr. 7A’ was changed to ‘T7AS-7S#1S-7AS.7AL’, which was labeled by Dubcovsky et al. (1998).

3. In Figure 4, T1CS401-2 (1 copy) displayed stronger HR compared to T1CS401-1 (3 copies). Similarly, in Supplemental Figure 11, T1CS401-7 (1 copies) displayed stronger HR compared to T1CS401-10 (5 copies). Why did such difference occurred? Is there difference in gene expression level among different lines with varied insertion copies?

Answer: The more effective resistance in the transgenic families T1CS401-1 and T1CS401-10 than in families T1CS401-2 and T1CS401-7 reflected the higher number of resistance genes present in them (or higher expression levels). Similar results have been reported in our previous studies, such as stem rust resistance genes *Sr21* (Chen et al. 2018; <https://doi.org/10.1371/journal.pgen.1007287>) and *Sr13* (Zhang et al. 2017; <https://doi.org/10.1073/pnas.1706277114>).

In this study, we described in the text (lines 248-250): “Among the transgenic plants, we found plants with both lower and higher levels of resistance than Kern *Lr47* (Supplementary Figs. 11a and 12), suggesting that these transgenic plants may have different numbers of *CNL2* insertions or expression levels.”. Then, in the next paragraph, we performed experiments to study the copy number and expression of *CNL2* transgene in 11 randomly selected transgenic T₀ plants. As described in

lines263-264: “The estimated *CNL2* copy number based on TaqMan assays correlated significantly with the transcript levels in the selected transgenic plants ($R = 0.90$, $P < 0.001$).”. Moreover, we added the following sentence in the next paragraph (lines269-271): “The more effective resistance observed in transgenic families T1CS401-1 and T1CS401-10 reflected the higher number of *CNL2* insertions (or higher expression levels) present in them.”

4. In Figure 5b, the author indicated the 7A chromosome (painted in red) with introgression segment of 7S (painted in green) marked with yellow arrowheads. Another pair of chromosomes also showed both red and green signals, and one of which was near closely to the arrowed chromosome. The author should explain why this pair of chromosomes produced green signals?

Answer: Thanks for pointing it out. We added the following sentence describing the other pair of chromosomes with both red and green signals (Lines341-342): “Kern *Lr47* also carries the previously reported 4AL/7BS intergenomic translocation (Fig. 5b), which is present in both durum and bread wheat²⁸.”

5. In addition, *ph1b* could induce recombination between 7S and 7A, it could also induce recombination between other homoeologous chromosomes of wheat. It was obvious to find large segment of translocation between D and A chromosomes in the Figure 5d. The author should analyze the structure variations of all the involved chromosomes including wheat chromosomes.

Answer: We agree with reviewer #2 that *ph1b* could also induce recombination between other homoeologous chromosomes of wheat, but it has not been the focus of our research. To bring it up, we added the following sentence describing recombination between other homoeologous chromosomes of wheat (lines343-345): “Moreover, we observed that the *ph1b* mutant induced additional recombination between the homoeologues of the A, B, and D subgenomes (Fig. 5d). Those translocations will be eliminated when the reduced introgressed segment is backcrossed into the target commercial wheat varieties.”

6. In Supplementary Fig. 10, all of the transgenic plants showed high resistance to *Pt*. Why did some of the leaves show strong HR while some leaves show no HR as observed in Kern *Lr47*?

Answer: The level of resistance in transgenic plants is associated with the number of *CNL2* insertions or expression levels. We explained the potential reason in the text (lines248-250): “Among the transgenic plants, we found plants with both lower and higher levels of resistance than Kern *Lr47* (Supplementary Figs. 11a and 12), suggesting that these transgenic plants may have different numbers of *CNL2* insertions or expression levels.”

In subsequent experiments (lines251-264), we found that the levels of resistance to *Pt* races in transgenic plants was significantly correlated with the copy number of *CNL2* or the expression levels. We added the following sentence in the next paragraph (lines269-271): “The more effective resistance observed in transgenic families T1CS401-1 and T1CS401-10 reflected the higher number of *CNL2* insertions (or higher expression levels) present in them.”

7. In Supplemental Figure 21, the scale bar was not clear.

Answer: We added the scale bars in the figure. In addition, we added the following information in the legend of Supplemental Figure 21 (now Supplemental Figure 23): “Scale bars represent 10 μm .”

8. In line 361, 'indicating that Lr47 is not induced by the presence of the Pt pathogen'. I suggested the author provided the leaves of Kern Lr47 with Pt inoculation and without Pt inoculation to compare the HR reaction before and after Pt infection.

Answer: We carried out the requested experiment. The leaves of Kern Lr47 with Pt inoculation and without Pt inoculation at five time points (0h, 1, 2, 4, and 6 dpi) are shown in the following figure. However, since Kern Lr47 confers very strong resistance and the sampling times still in early stages (pathogen are still growing), we did not observe obvious HR responses on leaves between two groups. Therefore, we didn't include this figure in the manuscript. In fact, the leaves of Kern Lr47 with Pt inoculation at 2-, 4-, 6-, and 8-dpi are displayed in Figure 1a.

9. A total of 23 Pt has been tested for their virulence to Lr47. Which type of the races are the currently virulent races threatening wheat production? I suggested the author supply the data about Pt races used in the manuscript to help breeders evaluating the value of Lr47 in current wheat breeding.

Answer: In the "Methods" section, we added the following sentence describing the predominant races threatening wheat production (lines 513-515): "The leaf rust races THT-, PHT-, THJ-, and PHJ- were predominant in China and showed virulence to most of the designated Lr genes^{43,44}". In addition, Avirulence/virulence formulae of Pt races used in this study has already been presented in Supplementary Table 1. Lr47 provides broad-spectrum resistance against all Pt races tested. In addition, we added the following information in the "Discussion" section: "Leaf rust isolates virulent on Lr47 have not been identified (line 479)". Therefore, Lr47 can be a valuable component of gene pyramids or transgenic cassettes combining different resistance genes to control this devastating disease.

10. Lr47 was characterized to be an NLR as a lot of cloned resistance genes from wheat and its relatives. In this study, there is little resistance mechanism revealed about Lr47. I suggest it's better to provide some special mechanism about the broad-spectrum resistance of Lr47.

Answer: We agree with the reviewer that we still do not have a mechanism for the broad resistance of Lr47, so we eliminated the "broad-spectrum" claim from the title and the text. In the section "Characterization of Lr47 encoding an NLR immune receptor protein", we have made efforts to provide an entry point to understand the mechanism of Lr47. Our results indicate that Lr47 has cytoplasmic and nuclear subcellular localization, and its coiled-coil (CC) domain was not able to induce cell death, nor did it have self-protein interaction. Further dissection of the mechanism is in progress, and may take a few years to complete and will be sufficient for a major follow-on paper when completed.

The main contribution of this manuscript is its societal relevance and the provision of a genetic tool to control leaf rust disease in wheat. Leaf rust causes significant yield losses annually in most wheat growing regions, but the resistance genes cloned so far are insufficient to provide adequate resistance to this rapidly evolving rust pathogen. A more diverse set of cloned resistance genes will be required to establish successful and durable combinations of multiple resistance genes in transgenic cassettes or to accelerate the deployment of diverse resistance gene-pyramids by marker-assisted selection. Cloning of the highly effective leaf rust resistance gene *Lr47* introgressed from *Aegilops speltoides* and the development of new *Lr47* introgression lines with no or reduced linkage drag still requires multiple years of work and technical expertise in several areas. Therefore, we believe that publication of such studies in high-impact journals such as Nature Communications remains important to encourage more research groups to invest their resources in solving problems in crop species that are relevant to our future global food security.

Reviewer #3:

Wheat leaf rust is one of the most severe foliar diseases in wheat production worldwide. Lr47 derived from a wild relative of wheat, Aegilops speltoides, provides broad-spectrum resistance. Lr47 was cloned by a combination of map-based cloning and EMS mutagenesis and transcript assembly (EMTA) approaches. Lr47 encodes a CC NLR protein, and was confirmed by mutation and transformation studies. From the practical point of view, the authors developed lines with smaller alien introgression, which did not have linkage drag. Markers were developed to be used in marker-assisted selection. In general, the manuscript was well written. Even though the gene cloned has a structure of an NLR-type, the findings in this paper have advanced our knowledge from both the fundamental and applied aspects. Hopefully the line with small introgression will be deployed to provide additional source for resistance breeding! Here are additional points for revision.

Answer: Thank you very much for recognition of our work.

Either in the “Abstract” or “Introduction” or both, it will be good to mention that Lr47 is a seedling / all-stage resistance gene, for the interest of readers.

Answer: We have added the word “all-stage” in the “Introduction” section (line68).

It will be good if the authors can add in the “Introduction” whether any virulence to Lr47 has been found anywhere in the world.

Answer: We believed that there was no virulence to *Lr47* in the world. In the “Introduction” section, we described (lines84-85): “This is a worthwhile endeavor because *Lr47* is one of a few genes known to confer strong levels of resistance against a wide range of *Pt* isolates^{1,13,17,19-23}”

Moreover, in the Discussion section, we added the following sentence (line479): “Leaf rust isolates virulent on *Lr47* have not been identified.”.

L70: Better use “homoeologous1”.

Answer: Corrected as suggested.

L244-247: What are the reasons resulting in the deviation from the 3:1 segregation ratio? What might be the reasons for preferential transmission, segregation distortion?

Answer: Because these transgenic families with more than one copy of *CNL2* insertions. It is normal that the transgenic plants have more than one independent functional insertions, as reported in cloning of rust resistance genes like Sr21 (Chen et al. 2018; <https://doi.org/10.1371/journal.pgen.1007287>) and Sr60 (Chen et al 2020; <https://doi.org/10.1111/nph.16169>). We explained the reason in the text as follows (lines260-263): “Overall, plants derived from three of the transgenic events (T₁CS401-2, T₁CS401-7, and T₁CS401-8) were estimated to have only a single copy of the transgene, whereas the other eight transgenic families were estimated to have between two and five *CNL2* copies (Supplementary Table 7).”

L268: Should be “longissima”; “83.29-92.78%”.

Answer: Corrected as suggested.

L285: Change “that” to “which”.

Answer: Corrected as suggested.

L355: Spell out “SNS, PH, TN” because they were not mentioned before.

Answer: We deleted this sentence from the manuscript.

L366: Use “Nicotiana” here and change to “N.” in L649.

Answer: Corrected as suggested.

L433: To follow the current taxonomy, better use “*Thinopyrum ponticum*” instead of “*Ag. elongatum*”.

Answer: Corrected as suggested.

L473: Change to “Lr34/Sr57/Yr18 and Lr67/Sr55/Yr46”.

Answer: Corrected as suggested.

L510: How many seeds were treated?

Answer: We added the following information in the “Methods” section (line529): “Approximately 20,000 seeds of wheat line Kern Lr47 were treated with 250 mL of 0.8% EMS...”

L607: The two references “75 and 76” cited did not have method for GISH, better add or change.

Answer: Thanks for pointing it out. We deleted references 75 and 76, and cited a new reference.

L610: oligo-pTa535 also hybridized to A-genome chromosomes in addition to D-genome ones.

Answer: Corrected as suggested.

L612-613: Need to explain better, e.g. “probe oligo-pTa713 generates additional diagnostic signals...” However, in the images (Fig. 5e, f, g), yellow signals of pTa-713 were almost invisible. In general, the FISH signals were not good, too weak. Fig. 5e and f are acceptable, but not Fig. 5d and g, which need a better cell.

Answer: Probe oligo-pTa713 was used for the identification of chromosome 7A. We rephrased the sentences as following (lines642-643): “Probe oligo-pTa713 generates additional diagnostic signals in the centromeric region of chromosome 7A⁶⁵.”

As suggested by the reviewer, we performed the requested experiments and obtained a better cell (as new Figure 5d and 5g). In addition, we have provided more GISH and FISH images as in new Supplementary Figure 21. The images were also adjusted to a suitable brightness and contrast.

Fig. 2c: It will be good to add the diagram of L8 here for comparison.

Answer: We respectfully disagree with the reviewer here. Because L8 was obtained from recombinant L2 after a second round of recombination with the *ph1b* mutant. Fig. 2c showed recombinant haplotypes when we performed fine mapping. If added, we will describe Fig. 2c in the section “Reducing the length of the introgressed segment carrying *Lr47*”. It would be good to describe the figures in order.

Fig. 4d and Supplementary Fig. 11: What’s the reason for the susceptible (having 1 copy of transgene) plants in T1CS401-2 and T1CS401-7?

Answer: Because transgenic families T₁CS401-2 and T₁CS401-7 with one single copy of transgene. Resistance in these two T₁ transgenic families fits the expected 3 resistant :1 susceptible segregation ratio (plants without the transgene were susceptible against *Pt*).

We added the following information in the figure and legend: “+, with transgene; -, without transgene.”

Fig. 6s: The spike length difference is not obvious. I am wondering whether it is better not to include this figure.

Answer: Agree. We deleted the figure 6s.

In the figure legend for Fig. 7e and Supplementary Fig. 22: It is better to add “No cell death was observed for”.

Answer: Corrected. We added the following sentences in the legends: “No cell death was observed in leaf regions transiently overexpressing *Lr47* and its protein domains individually. CC, coiled-coil; NB, nucleotide binding; LRR, leucine-rich repeat; CDS, the coding regions of *Lr47*.”

Supplementary Fig. 19: No need to show (a) and (b) because they did not provide much information.

Answer: Agree. We deleted (a) and (b) from Supplementary Fig. 19.

Supplementary Fig. 20: Not necessary as a figure. Suggest delete.

Answer: We deleted as suggested.

Supplementary Table 3: The avr/vir profiles are missing for the last four pathotypes.

Answer: Thanks for pointing it out. We added the avr/vir profiles for the last four pathotypes.

Reviewer #4:

*This study made tremendous efforts to clone a leaf rust gene, *Lr47*, in wheat. *Lr47* represents one of a few effective leaf rust resistance genes in wheat and the chromosomal segment harboring *Lr47* was transferred from *Aegilops speltoides*. Using two mapping wheat populations, the gene was mapped to a certain region including multiple NLR genes. Transcriptomes of EMS mutants were generated and assembled to obtain transcript sequences. Combination of mapping data, the reference genome, assembled transcripts, and potential EMS mutations led to a candidate gene, which was verified by sequence confirmation of EMS mutants, CRISPR knockouts through virus-mediated mutagenesis, and stable*

ectopic expression. The linkage drag introduced by the translocation segment was reduced by shortening the translocation segment through homoeologous recombination in the mutant of the *ph1* gene. In addition, a diagnostic molecular marker was developed to identify the potential *Lr47* donor, which can be also used to facilitate the deployment of the gene in breeding programs. Overall, the manuscript was well written and provided solid data for genetic mapping and candidate validation. I do not have major concerns regarding to the experimental design or data interpretation. I have the following minor comments for authors' references.

Answer: Thank you so much for your positive review of our manuscript.

1. The ethyl methanesulfonate (EMS) mutagenesis and transcript assembly (EMTA) approach was highlighted in the manuscript. In my opinion, this is a fairly regular and straightforward approach. It is not necessary to be highlighted.

Answer: We agree with the Reviewer's suggestion. We have deleted "EMS mutagenesis and transcript assembly (EMTA)" from the manuscript and cited the previous paper about cloning of the stem rust resistance gene *Sr62* (Yu et al., 2022).

However, compared with the previously reported MutRNASeq approach, there are differences between our method and the previous MutRNASeq method. In our Methods section (lines 552-558), we described the procedure of our method: "The modified MutRNASeq was performed as follows: RNA-seq reads from Kern *Lr47* were assembled *de novo* using Spades version 3.14.1⁵⁴. We performed BLASTN searches of the transcriptome database of Kern *Lr47* using sequences of typical NLR genes within the *Lr47* candidate regions from the reference genomes of CS and TS01⁵⁵ as queries, and obtained transcript contigs with similarity values greater than 85% to the homologs or paralogs within the interval that *Lr47* was mapped to. Next, the RNA-seq reads of the ten susceptible EMS mutants were mapped to these transcript contigs as reported previously²⁴."

The key difference is that our method "does not require the generation of a physical map/reference sequence across the map interval (line416)". Since the mapping candidate regions in TS01 and CS reference genomes have a cluster of NLR genes, we hypothesized that *Lr47* might be an NLR genes, and then we performed the following analysis -- as described in the Results section (lines190-196): "Assembly of the Kern *Lr47* sequenced reads yielded 146,715 high confidence transcript contigs (\geq 500 bp). We performed BLASTN searches of the Kern *Lr47* transcriptome database using the sequences of the candidate NLR genes in CS and TS01 as queries, and obtained 45 transcript contigs with a BLAST e-value = 0. We mapped the sequenced reads of ten susceptible mutants against the 45 selected Kern *Lr47* transcript contigs and found one contig named KN638873_g379_i12 with EMS-type (G/C-to-A/T) point mutations in all ten mutants (Supplementary Figs. 6b and 7a)." Thus, we would like to term this as "modified MutRNASeq" in the new version R1.

2. L75: when chromosomal coordinates are described, it would be informative if the reference genome (species and genome version) is specified.

Answer: Agree. We rephrased the sentences as follows (line74-76): "In a recent study, the length of the *Ae. speltoides* segment 7S#1S was estimated to be between 157 and 174 Mb (Chinese Spring RefSeq v1.0 coordinates) based on a set of simple sequence repeat (SSR) markers."

3. L129: I might miss something, But I was confused when I read the paragraph from L118-130. How was the following conclusion drawn. "... Thus, the *Ae. speltoides* segment in the three *Lr47* NILs ranges from 150.1 Mb to 151.8 Mb". Sorry, I just did not see the logic.

Answer: Using RNA Sequencing + SNP calling + bioinformatic analysis, we focused on the SNPs that were present in the three *Ae. speltooides* accessions (1-3 lines in Supplementary Fig. 2), but absent in the 10 sequenced hexaploid wheat varieties (7-16 lines in Supplementary Fig. 2), which are referred to as *Ae. speltooides*-specific SNPs. Based on this rule, we identified 3,169 *Ae. speltooides*-specific SNPs (vertical lines in blue; Supplementary Data1) that were shared with *Lr47* NILs in the proximal region of chromosome arm 7AS starting from 40.4 Mb to 190.5 Mb (CS RefSeq v1.1 coordinates). Thus, we were able to determinate the *Ae. speltooides* segment in the three *Lr47* NILs (4-6 lines in Supplementary Fig. 2) ranges from 150.1 Mb to 151.8 Mb (see the following Supplementary Figure 2).

In addition, we designed 16 7A/7S genome-specific primers across the ~150 Mb introgressed *Ae. speltooides* segment (Supplementary Table 3), and used them to genotype the *Lr47* NILs and their recurrent parents. We found that translocation breakpoints in the *Lr47* NILs were located between markers *pku0738* and *pku0745* at the terminal end, and between markers *pku2216* and *pku2233* at the proximal end (Supplementary Fig. 3).

Supplementary Fig. 2. Distribution of 3,523 putative *Ae. speltooides*-specific SNPs on chromosome 7AS. The introgressed *Ae. speltooides* segment in Kern *Lr47*, Yecora Rojo *Lr47*, and UC1041 *Lr47* was approximately 150 Mb long, extending from 40 Mb to 190 Mb. 1-3) *Ae. speltooides* accessions AE915, AE1590, and PI 554292; 4-6) *Lr47* introgression lines Kern *Lr47*, Yecora Rojo *Lr47*, and UC1041 *Lr47*; 7-16) Sequenced *T. aestivum* accessions Norin61, CDC Stanley, Mace, Julius, Arina *LrFor*, Jagger, LongReach Lancer, CDC LandMark, SY Mattis, and Chinese Spring (Supplementary Data 1). Integrative Genomics Viewer (IGV) software version 2.8.9 was used to visualize the distribution of these SNPs. Vertical lines in blue indicate *Ae. speltooides*-specific SNPs, while lines in light grey represent normal wheat SNPs. Coordinates are based on CS RefSeq v1.1.

4. L161: When m118 was first mentioned, the EMS experiment has not described. It would be clearer to mention that the EMS experiment was performed and one of the mutants was used to build a mapping population.

Answer: Thank you for pointing this out. To resolve the problem, we moved this paragraph to the next section, and described the m118 x Kern *Lr47* (WT) mapping population after we obtained susceptible EMS lines.

We rephrased the sentence as follows (lines 176-178): “we crossed one of the obtained susceptible EMS mutant lines, m118, with the non-mutagenized resistant Kern *Lr47*, and generated an F₂ population consisting of 1,141 individuals.”

5. L202: To be clear, “the mutations” can be replaced by “the EMS mutations”.

Answer: Corrected as suggested.

6. L376: It would be helpful to briefly describe the purpose of BAX and Sr13 in “expressing BAX and the CC domain of Sr13”.

Answer: We apologize for not giving a clear description. We rephrased the sentences as following (lines 393-395): “In contrast, robust cell death was observed in leaf regions expressing BAX (an inducer of cell death)²⁹ and the CC domain of Sr13³⁰, which were used as positive controls (Fig. 7e and Supplementary Fig. 24).”.

7. The abbreviation of the first appearance of a term should be spelled out and followed by its abbreviation in parentheses, such as Genomic In Situ Hybridization (GISH). I saw the full name in the Methods section, which is located behind the Results section, it is appropriate to use the abbreviation thereafter.

Answer: Corrected as suggested.

In addition to the reviewer’s suggested changes, we deleted the previous “Supplementary Fig. 7” and relevant description from the manuscript. Moreover, we deleted the following paragraphs from the Discussion section:

“However, the effectiveness of the EMTA method relies on several factors: (1) the delimitation of the target gene within a physical interval; (2) the isolation of multiple independent mutants carrying loss-of-function mutations; (3) the presence of orthologs or homologs or paralogs in the colinear regions in sequenced reference genomes; and (4) the comparison of the selected mutants with wild type using RNA-seq data.”

“Sequence alignments of the *Lr47* protein and representative homologs revealed critical amino acid polymorphisms, which we used to develop a diagnostic marker for the presence of *Lr47* (Supplementary Figs. 14 and 15). We validated this diagnostic marker in a larger number of accessions of *T. monococcum*, *T. turgidum*, *T. aestivum*, and *Ae. speltoides* (Supplementary Table 10). We detected *Lr47* in only 2.5% of the *Ae. speltoides* genotypes and in none of the diploid, tetraploid, and hexaploid wheat accessions tested, except for six *Lr47* NILs. This result indicates that the incorporation of *Lr47* has the potential to improve leaf rust resistance in a wide range of modern wheat varieties and highlights the importance of mining new *R* genes from wild progenitors.”

Reviewers' Comments:

Reviewer #1:

Remarks to the Author:

We have carefully evaluated the rebuttal and revised manuscript.

Regarding our main concern, namely the claim for 'broad-spectrum resistance' of Lr47 in the original manuscript, the authors removed this terminology from the revised manuscript. In addition, they did an additional experiment in which they inoculated their transgenic and or EMS mutant plants with seven defined leaf rust isolates as well as a field mix (all from China, if we are not mistaken). They did not detect any virulence on the transgenics, and the mutants retained susceptibility.

There is one caveat with these experiments - without knowing the genetic relatedness of these isolates it is difficult to strengthen any claims about 'broad-spectrum resistance' because, for all we know, the different races could have been derived through stepwise gains in virulence. Therefore, we feel the authors made the correct call in removing the claim for 'broad-spectrum resistance' and replacing it with the terminology "broadly effective". We agree with this revision.

In conclusion, it was a great pleasure to review this fine manuscript and careful revision. We salute the authors for their thorough work.

Reviewer #2:

Remarks to the Author:

Identification of Lr resistance genes to multiple Pt isolates shows great value in wheat breeding, and wild species are valuable resources for mining such kind of resistance. In this study, the Pt resistance gene Lr47, identified from *Aegilops speltoides*, was cloned by fine mapping, mutants creation, gene knockout and stable transformation. The manuscript provided strong evidence to support the conclusion that NLR2 is the Lr47 gene. Moreover, the germplasm with small introgression segment was also identified for wheat breeding. In the revised manuscript, the comments from 4 reviewers were well answered, and new data were provided as requested. I think the quality of the revised manuscript has been greatly improved and I recommend for its publication in Nature Communication.

Reviewer #3:

Remarks to the Author:

The manuscript was further improved a lot after incorporating the suggestions/comments from all the reviewers. I am happy with the responses that the authors provided and the revised manuscript.

A minor technical point: In Line 529, 250 mL of solution would be barely enough to immerse 20,000 seeds. Please check.

Reviewer #4:

Remarks to the Author:

The authors have well addressed my questions/comments. I have no further questions.

Reviewer #1:

We have carefully evaluated the rebuttal and revised manuscript.

Regarding our main concern, namely the claim for 'broad-spectrum resistance' of Lr47 in the original manuscript, the authors removed this terminology from the revised manuscript. In addition, they did an additional experiment in which they inoculated their transgenic and or EMS mutant plants with seven defined leaf rust isolates as well as a field mix (all from China, if we are not mistaken). They did not detect any virulence on the transgenics, and the mutants retained susceptibility.

There is one caveat with these experiments - without knowing the genetic relatedness of these isolates it is difficult to strengthen any claims about 'broad-spectrum resistance' because, for all we know, the different races could have been derived through stepwise gains in virulence. Therefore, we feel the authors made the correct call in removing the claim for 'broad-spectrum resistance' and replacing it with the terminology "broadly effective". We agree with this revision.

In conclusion, it was a great pleasure to review this fine manuscript and careful revision. We salute the authors for their thorough work.

Author's Answer: We thank the Reviewer for the encouraging comments.

Reviewer #2:

*Identification of Lr resistance genes to multiple Pt isolates shows great value in wheat breeding, and wild species are valuable resources for mining such kind of resistance. In this study, the Pt resistance gene Lr47, identified from *Aegilops speltoides*, was cloned by fine mapping, mutants creation, gene knockout and stable transformation. The manuscript provided strong evidence to support the conclusion that NLR2 is the Lr47 gene. Moreover, the germplasm with small introgression segment was also identified for wheat breeding. In the revised manuscript, the comments from 4 reviewers were well answered, and new data were provided as requested. I think the quality of the revised manuscript has been greatly improved and I recommend for its publication in Nature Communication.*

Author's Answer: We thank the Reviewer for the encouraging comments.

Reviewer #3:

The manuscript was further improved a lot after incorporating the suggestions/comments from all the reviewers. I am happy with the responses that the authors provided and the revised manuscript.

A minor technical point: In Line 529, 250 mL of solution would be barely enough to immerse 20,000 seeds. Please check.

Author's Answer: Thanks for pointing it out. In this experiment, we treated approximately 20,000 seeds of the wheat line Kern Lr47. The seeds were placed in 16 separate beakers, each containing 250 mL of a 0.8% EMS solution (Cat. No. M0880-25G, Sigma-Aldrich, USA). These beakers were then incubated for 18 hours at a temperature of 25 °C while shaking at a speed of 150 rpm. The sentence was rephrased as follow: "Approximately 20,000 seeds of wheat line Kern Lr47 were treated in 16 beakers, each containing 250 mL of 0.8% EMS (Cat No. M0880-25G, Sigma-Aldrich, USA). The beakers were incubated for 18 h at 25 °C on a shaker at 150 rpm."

Reviewer #4:

The authors have well addressed my questions/comments. I have no further questions.

Author's Answer: We thank the Reviewer for the encouraging comments.